# Fairness Aware Counterfactuals for Subgroups

**Loukas Kavouras**
IMSI / Athena RC
kavouras@athenarc.gr

**Konstantinos Tsopelas**
IMSI / Athena RC
k.tsopelas@athenarc.gr

**Giorgos Giannopoulos**
IMSI / Athena RC
giann@athenarc.gr

**Dimitris Sacharidis**
ULB
dimitris.sacharidis@ulb.be

**Eleni Psaroudaki**
NTUA & IMSI / Athena RC
epsaroudaki@mail.ntua.gr

**Nikolaos Theologitis**
IMSI / Athena RC
n.theologitis@athenarc.gr

**Dimitrios Rontogiannis**
IMSI / Athena RC
dronto@gmail.com

**Dimitris Fotakis**
NTUA & Archimedes / Athena RC
fotakis@cs.ntua.gr

**Ioannis Emiris**
NKUA & IMSI / Athena RC
emiris@athenarc.gr

## Abstract

In this work, we present Fairness Aware Counterfactuals for Subgroups (FACTS), a framework for auditing subgroup fairness through counterfactual explanations. We start with revisiting (and generalizing) existing notions and introducing new, more refined notions of subgroup fairness. We aim to (a) formulate different aspects of the difficulty of individuals in certain subgroups to achieve recourse, i.e., receive the desired outcome, either at the micro level, considering members of the subgroup individually, or at the macro level, considering the subgroup as a whole, and (b) introduce notions of subgroup fairness that are robust, if not totally oblivious, to the cost of achieving recourse. We accompany these notions with an efficient, model-agnostic, highly parameterizable, and explainable framework for evaluating subgroup fairness. We demonstrate the advantages, the wide applicability, and the efficiency of our approach through a thorough experimental evaluation of different benchmark datasets.

## 1 Introduction

Machine learning is now an integral part of decision-making processes across various domains, e.g., medical applications [22], employment [5, 7], recommender systems [29], education [26], credit assessment [4]. Its decisions affect our everyday life directly, and, if unjust or discriminative, could potentially harm our society [28]. Multiple examples of discrimination or bias towards specific population subgroups in such applications [33] create the need not only for explainable and interpretable machine learning that is more trustworthy [6], but also for auditing models in order to detect hidden bias for subgroups [30].

Bias towards protected subgroups is most often detected by various notions of *fairness of prediction*, e.g., statistical parity, where all subgroups defined by a protected attribute should have the same probability of being assigned the positive (favorable) predicted class. These definitions capture the *explicit* bias reflected in the model's predictions. Nevertheless, an *implicit* form of bias is the

37th Conference on Neural Information Processing Systems (NeurIPS 2023).

difficulty for, or the *burden* [32, 20] of, an individual (or a group thereof) to achieve *recourse*, i.e., perform the necessary *actions* to change their features so as to obtain the favorable outcome [10, 35]. Recourse provides explainability (i.e., a counterfactual explanation [38]) and actionability to an affected individual, and is a legal necessity in various domains, e.g., the Equal Credit Opportunity Act mandates that an individual can demand to learn the reasons for a loan denial. *Fairness of recourse* captures the notion that the protected subgroups should bear equal burden [10, 14, 37, 20].

To illustrate these notions, consider a company that supports its promotion decisions with an AI system that classifies employees as good candidates for promotion, the favorable positive class, based on various performance metrics, including their cycle time efficiency (CTE) and the annual contract value (ACV) for the projects they lead. Figure 1a draws ten employees from the negative predicted class as points in the CTE-ACV plane and also depicts the decision boundary of the classifier. Race is the protected attribute, and there are two protected subgroups with five employees each, depicted as circles and triangles. For each employee, the arrow depicts the best *action* to achieve recourse, i.e., to cross the decision boundary, and the number indicates the *cost* of the action, here simply computed as the distance to the boundary [10]. For example, $x_1$ may increase their chances for promotion mostly by acquiring more high-valued projects, while $x_2$ mostly by increasing their efficiency. Burden is defined as the mean cost for a protected subgroup [32]. For the protected race 0, the burden is 2, while it is 2.2 for race 1, indicating thus unfairness of recourse against race 1. In contrast, assuming there is an equal number of employees of each race in the company, the classifier satisfies fairness of prediction in terms of statistical parity (equal positive rate in the subgroups).

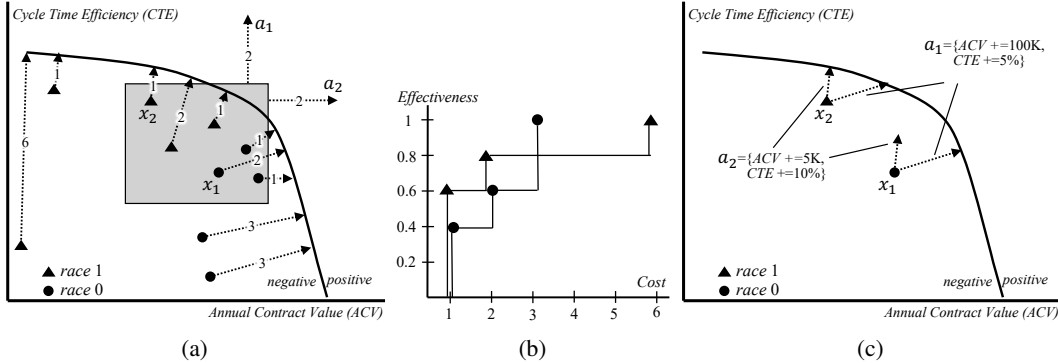

Figure 1: (a) An example of affected individuals, the decision boundary, actions, and a subpopulation (in the shaded region), depicted in the feature space; (b) Cumulative distribution of cost of recourse for the individuals in (a); (c) Comparison of two actions to achieve recourse for two individuals.

While fairness of recourse is an important concept that captures a distinct notion of algorithmic bias, as also explained in [14, 37], we argue that it is much more nuanced than the mean cost of recourse (aka burden) considered in all prior work [10, 32, 37, 20], and raise three issues.

First, the mean cost, which comprises a *micro viewpoint* of the problem, does not provide the complete picture of how the cost of recourse varies among individuals and may lead to contrasting conclusions. Consider again Figure 1a, and observe that for race 1 all but one employee can achieve recourse with cost at most 2, while an outlier achieves recourse with cost 6. It is this outlier that raises the mean cost for race 1 above that of race 0. For the two race subgroups, Figure 1b shows the cumulative distribution of cost, termed the *effectiveness-cost distribution* (ecd) in Section 2. These distributions allow the fairness auditor to inspect the *tradeoff* between cost and recourse, and define the appropriate fairness of recourse notion. For example, they may consider that actions with a cost of more than 2 are unrealistic (e.g., because they cannot be realized within some timeframe), and thus investigate how many employees can achieve recourse under this constraint; we refer to this as *equal effectiveness within budget* in Section 2.3. Under this notion, there is unfairness against race 0, as only 60% of race 0 employees (compared to 80% of race 1) can realistically achieve recourse.

There are several options to go beyond the mean cost. One is to consider fairness of recourse at the individual level, and compare an individual with their counterfactual counterpart had their protected attribute changed value [37]. However, this approach is impractical as, similar to other causal-based definitions of fairness, e.g., [21], it requires strong assumptions about the causal structure in the

domain [15]. In contrast, we argue that it's preferable to investigate fairness in subpopulations and inspect the trade-off between cost and recourse.

Second, there are many cases where the aforementioned micro-level aggregation of individuals' costs is not meaningful in auditing real-world systems. To account for this, we introduce a *macro viewpoint* where a group of individuals is considered as a whole, and an action is applied to and assessed *collectively* for all individuals in the group. An action represents an external horizontal intervention, such as an affirmative action in society, or focused measures in an organization that would change the attributes of some subpopulation (e.g., decrease tax or loan interest rates, increase productivity skills). In the macro viewpoint, the cost of recourse does not burden the individuals, but the external third party, e.g., the society or an organization. Moreover, the macro viewpoint offers a more intuitive way to *audit* a system for fairness of recourse, as it seeks to uncover systemic biases that apply to a large number of individuals.

To illustrate the macro viewpoint, consider the group within the shaded region in Figure 1a. In the micro viewpoint, each employee seeks recourse individually, and both race subgroups have the same distribution of costs. However, we can observe that race 0 employees, like $x_1$ achieve recourse by actions in the ACV direction, while race 1 employees, like $x_2$ in the orthogonal CTE direction. This becomes apparent when we take the macro viewpoint, and investigate the effect of action $a_1$, depicted on the border of the shaded region, discovering that it is disproportionally effective on the race subgroups (leads to recourse for two-thirds of one subgroup but for none in the other). In this example, action $a_1$ might represent the effect of a training program to enhance productivity skills, and the macro viewpoint finds that it would perpetuate the existing burden of race 0 employees.

Third, existing notions of fairness of recourse have an important practical limitation: they require a cost function that captures one's ability to modify one's attributes, whose definition may involve a learning process [31], or an adaptation of off-the-shelf functions by practitioners [35]. Even the idea of which attributes are actionable, in the sense that one can change (e.g., one cannot get younger), or "ethical" to suggest as actionable (e.g., change of marital status from married to divorced) hides many complications [36].

Conclusions drawn about fairness of recourse crucially depend on the cost definition. Consider individuals $x_1$, $x_2$, and actions $a_1$, $a_2$, shown in Figure 1c. Observe that is hard to say which action is cheaper, as one needs to compare changes *within* and *across* very dissimilar attributes. Suppose the cost function indicates action $a_1$ is cheaper; both individuals achieve recourse with the same cost. However, if action $a_2$ is cheaper, only $x_2$ achieves recourse. Is the classifier fair?

To address this limitation, we propose definitions that are *oblivious to the cost function*. The idea is to compare the effectiveness of actions to the protected subgroups, rather than the cost of recourse for the subgroups. One way to define a cost-oblivious notion of fairness of recourse is the *equal choice for recourse* (see Section 2.3), which we illustrate using the example in Figure 1c. According to it, the classifier is unfair against $x_1$, as $x_1$ has only one option, while $x_2$ has two options to achieve recourse among the set of actions $\{a_1, a_2\}$.

**Contribution**   Our aim is to showcase that fairness of recourse is an important and distinct notion of algorithmic fairness with several facets not previously explored. We make a series of conceptual and technical contributions.

Conceptually, we distinguish between two different viewpoints. The *micro viewpoint* follows literature on recourse fairness [10, 14, 37, 20] in that each individual chooses the action that is cheaper for them, but revisits existing notions. It considers the trade-off between cost and recourse and defines several novel notions that capture different aspects of it. The *macro viewpoint* considers how an action collectively affects a group of individuals, and quantifies its effectiveness. It allows the formulation of *cost-oblivious* notions of recourse fairness. It also leads to an alternative trade-off between cost and recourse that may reveal systemic forms of bias.

Technically, we propose an efficient, interpretable, model-agnostic, highly parametrizable framework termed FACTS (Fairness-Aware Countefactuals for Subgroups) to audit for fairness of recourse. FACTS is *efficient* in computing the effectiveness-cost distribution, which captures the trade-off between cost and recourse, in both the micro or macro viewpoint. The key idea is that instead of determining the best action for each individual independently (i.e., finding their nearest counterfactual explanation [38]), it enumerates the space of actions and determines how many and which individuals

achieve recourse through each action. Furthermore, FACTS employs a systematic way to explore the feature space and discover any subspace such that recourse bias exists among the protected subgroups within. FACTS ranks the subspaces in decreasing order of recourse bias it detects, and for each provides an *interpretable* summary of its findings.

**Related Work** We distinguish between *fairness of predictions* and *fairness of recourse*. The former aims to capture and quantify unfairness by comparing directly the model's predictions [27, 2] at: the individual level, e.g., individual fairness [9], counterfactual/causal-based fairness [21, 19, 26]; and the group level, e.g., demographic parity [39], equal odd/opportunity [12].

Fairness of recourse is a more recent notion, related to *counterfactual explanations* [38], which explains a prediction for an individual (the factual) by presenting the "best" counterfactual that would result in the opposite prediction, offering thus *recourse* to the individual. Best, typically means the *nearest counterfactual* in terms of a distance metric in the feature space. Another perspective, which we adopt here, is to consider the *action* that transforms a factual into a counterfactual, and specify a *cost function* to quantify the effort required by an individual to perform an action. In the simplest case, the cost function can be the distance between factual and counterfactual, but it can also encode the *feasibility* of an action (e.g., it is impossible to decrease age) and the *plausibility* of a counterfactual (e.g., it is out-of-distribution). It is also possible to view actions as interventions that act on a structural causal model capturing cause-effect relationships among attributes [15]. Hereafter, we adopt the most general definition, where a cost function is available, and assume that the best counterfactual explanation is the one that comes from the minimum cost action. Counterfactual explanations have been suggested as a mechanism to detect possible bias against protected subgroups, e.g., when they require changes in protected attributes [13].

Fairness of recourse, first introduced in [35] and formalized in [10], is defined at the group level as the disparity of the mean cost to achieve recourse (called burden in subsequent works) among the protected subgroups. Fairness of recourse for an individual is when they require the same cost to achieve recourse in the actual world and in an imaginary world where they would have a different value in the protected attribute [37]. This definition however only applies when a structural causal model of the world is available. Our work expands on these ideas and proposes alternate definitions that capture a macro and a micro viewpoint of fairness of recourse.

There is a line of work on auditing models for fairness of predictions at the subpopulation level [17, 18]. For example, [34] identifies subpopulations that show dependence between a performance measure and the protected attribute. [3] determines whether people are harmed due to their membership in a specific group by examining a ranking of features that are most associated with the model's behavior. There is no equivalent work for fairness of recourse, although the need to consider the subpopulation is recognized in [16] due to uncertainty in assumptions or to intentionally study fairness. Our work is the first that audits for fairness of recourse at the subpopulation level.

A final related line of work is global explainability. For example, recourse summaries [31, 24, 25] summarizes individual counterfactual explanations *globally*, and as the authors in [31, 25] suggest can be used to manually audit for unfairness in subgroups of interest. [23] aims to explain how a model behaves in subspaces characterized by certain features of interest. [8] uses counterfactuals to unveil whether a black-box model, that already complies with the regulations that demand the omission of sensitive attributes, is still biased or not, by trying to find a relation between proxy features and bias. Our work is related to these methods in that we also compute counterfactual explanations for all instances, albeit in a more efficient manner, and with the goal to audit fairness of recourse on subpopulation level.

## 2 Fairness of Recourse for Subgroups

### 2.1 Preliminaries

We consider a **feature space** $X = X_1 \times \cdots \times X_n$, where $X_n$ denotes the **protected feature**, which, for ease of presentation, takes two protected values $\{0, 1\}$. For an instance $x \in X$, we use the notation $x.X_i$ to refer to its value in feature $X_i$.

We consider a binary **classifier** $h : X \to \{-1, 1\}$ where the positive outcome is the favorable one. For a given $h$, we are concerned with a dataset $D$ of **adversely affected individuals**, i.e., those who

receive the unfavorable outcome. We prefer the term instance to refer to any point in the feature space $X$, and the term individual to refer to an instance from the dataset $D$.

We define an **action** $a$ as a set of changes to feature values, e.g., $a = \{country \rightarrow US, education\text{-}num \rightarrow 12\}$. We denote as $A$ the set of possible actions. An action $a$ when applied to an individual (a factual instance) $x$ results in a **counterfactual** instance $x' = a(x)$. If the individual $x$ was adversely affected ($h(x) = -1$) and the action results in a counterfactual that receives the desired outcome ($h(a(x)) = 1$), we say that action $a$ offers recourse to the individual $x$ and is thus **effective**. In line with the literature, we also refer to an effective action as a **counterfactual explanation** for individual $x$ [38].

An action $a$ incurs a **cost** to an individual $x$, which we denote as $\mathsf{cost}(a, x)$. The cost function captures both how *feasible* the action $a$ is for the individual $x$, and how *plausible* the counterfactual $a(x)$ is [14].

Given a set of actions $A$, we define the **recourse cost** $\mathsf{rc}(A, x)$ of an individual $x$ as the minimum cost among effective actions if there is one, or otherwise some maximum cost represented as $c_\infty$:

$$\mathsf{rc}(A, x) = \begin{cases} \min\{\mathsf{cost}(a, x) | a \in A : h(a(x)) = 1\}, & \text{if } \exists a \in A : h(a(x)) = 1; \\ c_\infty, & \text{otherwise.} \end{cases}$$

An effective action of minimum cost is also called a **nearest counterfactual explanation** [14].

We define a **subspace** $X_p \subseteq X$ using a **predicate** $p$, which is a conjunction of feature-level predicates of the form "*feature-operator-value*", e.g., the predicate $p = (country = US) \wedge (education\text{-}num \geq 9)$ defines instances from the US that have more than 9 years of education.

Given a predicate $p$, we define the subpopulation **group** $G_p \subseteq D$ as the set of affected individuals that satisfy $p$, i.e., $G_p = \{x \in D | p(x)\}$. We further distinguish between the **protected subgroups** $G_{p,1} = \{x \in D | p(x) \wedge x.X_n = 1\}$ and $G_{p,0} = \{x \in D | p(x) \wedge x.X_n = 0\}$. When the predicate $p$ is understood, we may omit it in the designation of a group to simplify notation.

## 2.2 Effectiveness-Cost Trade-Off

For a specific action $a$, we naturally define its **effectiveness** (eff) for a group $G$, as the proportion of individuals from $G$ that achieve recourse through $a$:

$$\mathsf{eff}(a, G) = \frac{1}{|G|} |\{x \in G | h(a(x)) = 1\}|.$$

Note that effectiveness is termed correctness in [31] and coverage in [25]. We want to examine how recourse is achieved for the group $G$ through a set of possible actions $A$. We define the **aggregate effectiveness** (aeff) of $A$ for $G$ in two distinct ways.

In the *micro viewpoint*, the individuals in the group are considered independently, and each may choose the action that benefits itself the most. Concretely, we define the **micro-effectiveness** of set of actions $A$ for group $G$ as the proportion of individuals in $G$ that can achieve recourse through *some* action in $A$, i.e.,:

$$\mathsf{aeff}_\mu(A, G) = \frac{1}{|G|} |\{x \in G | \exists a \in A, \mathsf{eff}(a, x) = 1\}|.$$

In the *macro viewpoint*, the group is considered as a whole, and an action is applied collectively to all individuals in the group. Concretely, we define the **macro-effectiveness** of set of actions $A$ for group $G$ as the largest proportion of individuals in $G$ that can achieve recourse through *the same* action in $A$, i.e.,:

$$\mathsf{aeff}_\mathsf{M}(A, G) = \max_{a \in A} \frac{1}{|G|} |\{x \in G | \mathsf{eff}(a, x) = 1\}|.$$

For a group $G$, actions $A$, and a cost budget $c$, we define the **in-budget actions** as the set of actions that cost at most $c$ for any individual in $G$:

$$A_c = \{a \in A | \forall x \in G, \mathsf{cost}(a, x) \leq c\}.$$

We define the **effectiveness-cost distribution** (ecd) as the function that for a cost budget $c$ returns the aggregate effectiveness possible with in-budget actions:

$$\mathsf{ecd}(c; A, G) = \mathsf{aeff}(A_c, G).$$

We use $\mathsf{ecd}_\mu$, $\mathsf{ecd}_\mathsf{M}$ to refer to the micro, macro viewpoints of aggregate effectiveness. A similar concept, termed the coverage-cost profile, appears in [25].

The value $\mathsf{ecd}(c; A, G)$ is the proportion of individuals in $G$ that can achieve recourse through actions $A$ with cost at most $c$. Therefore, the ecd function has an intuitive probabilistic interpretation. Consider the subspace $X_p$ determined by predicate $p$, and define the random variable $C$ as the cost required by an instance $x \in X_p$ to achieve recourse. The function $\mathsf{ecd}(c; A, G_p)$ is the empirical cumulative distribution function of $C$ using sample $G_p$.

The **inverse effectiveness-cost distribution** function $\mathsf{ecd}^{-1}(\phi; A, G)$ takes as input an effectiveness level $\phi \in [0, 1]$ and returns the minimum cost required so that $\phi|G|$ individuals achieve recourse.

## 2.3 Definitions of Subgroup Recourse Fairness

We define recourse fairness of classifier $h$ for a group $G$ by comparing the ecd functions of the protected subgroups $G_0$, $G_1$ in different ways.

The first two definitions are *cost-oblivious*, and apply whenever we can ignore the cost function. Specifically, given a set of actions $A$ and a group $G$, we assume that all actions in $A$ are considered equivalent, and that the cost of any action is the same for all individuals in the group; i.e., $\mathsf{cost}(a, x) = \mathsf{cost}(a', x'), \forall a \neq a' \in A, x \neq x' \in G$. The definitions simply compare the aggregate effectiveness of a set of actions on the protected subgroups.

**Equal Effectiveness** This definition has a micro and a macro interpretation, and says that the classifier is fair if the same proportion of individuals in the protected subgroups can achieve recourse: $\mathsf{aeff}(A, G_0) = \mathsf{aeff}(A, G_1)$.

**Equal Choice for Recourse** This definition has only a macro interpretation and claims that the classifier is fair if the protected subgroups can choose among the same number of sufficiently effective actions to achieve recourse, where sufficiently effective means the actions should work for at least $100\phi\%$ (for $\phi \in [0, 1]$) of the subgroup:

$$|\{a \in A \,|\, \mathsf{eff}(a, G_0) \geq \phi\}| = |\{a \in A \,|\, \mathsf{eff}(a, G_1) \geq \phi\}|.$$

The next three definitions assume the cost function is specified, and have both a micro and a macro interpretation.

**Equal Effectiveness within Budget** The classifier is fair if the same proportion of individuals in the protected subgroups can achieve recourse with a cost at most $c$:

$$\mathsf{ecd}(c; A, G_0) = \mathsf{ecd}(c; A, G_1).$$

**Equal Cost of Effectiveness** The classifier is fair if the minimum cost to achieve aggregate effectiveness of $\phi \in [0, 1]$ in the protected subgroups is equal:

$$\mathsf{ecd}^{-1}(\phi; A, G_0) = \mathsf{ecd}^{-1}(\phi; A, G_1).$$

**Fair Effectiveness-Cost Trade-Off** The classifier is fair if the protected subgroups have the same effectiveness-cost distribution, or equivalently for each cost budget $c$, their aggregate effectiveness is equal:

$$\max_c |\,\mathsf{ecd}(c; A, G_0) - \mathsf{ecd}(c; A, G_1)| = 0.$$

The left-hand side represents the two-sample Kolmogorov-Smirnov statistic for the empirical cumulative distributions (ecd) of the protected subgroups. We say that the classifier is fair with confidence $\alpha$ if this statistic is less than $\sqrt{-\ln(\alpha/2) \frac{|G_{p,0}| + |G_{p,1}|}{2|G_{p,0}||G_{p,1}|}}$.

The last definition takes a micro viewpoint and extends the notion of *burden* [32] from literature to the case where not all individuals may achieve recourse. The **mean recourse cost** of a group $G$,

$$\bar{\mathsf{rc}}(A, G) = \frac{1}{|G|} \sum_{x \in G} \mathsf{rc}(A, x)$$

considers individuals that cannot achieve recourse through $A$ and have a recourse cost of $c_\infty$. To exclude them, we denote as $G^*$ the set of individuals of $G$ that can achieve recourse through an action in $A$, i.e., $G^* = \{x \in G | \exists a \in A, h(a(x)) = 1\}$. Then the **conditional mean recourse cost** is the mean recourse cost among those that can achieve recourse:

$$\bar{\mathsf{rc}}^*(A, G) = \frac{1}{|G^*|} \sum_{x \in G^*} \mathsf{rc}(A, x).$$

If $G = G^*$, the definitions coincide with burden.

**Equal (Conditional) Mean Recourse** The classifier is fair if the (conditional) mean recourse cost for the protected subgroups is the same:

$$\bar{\mathsf{rc}}^*(A, G_0; A) = \bar{\mathsf{rc}}^*(A, G_1).$$

Note that when the group $G$ is the entire dataset of affected individuals, and all individuals can achieve recourse through $A$, this fairness notion coincides with fairness of burden [35, 32].

## 3 Fairness-aware Counterfactuals for Subgroups

This section presents FACTS (Fairness-aware Counterfactuals for Subgroups), a framework that implements both the micro and the macro viewpoint, and all respective fairness definitions provided in Section 2.3 to support auditing of the "difficulty to achieve recourse" in subgroups. The output of FACTS comprises population groups that are assigned (a) an unfairness score that captures the disparity between protected subgroups according to different fairness definitions and allows us to rank groups, and (b) a user-intuitive, easily explainable counterfactual summary, which we term *Comparative Subgroup Counterfactuals* (CSC).

Figure 2 presents an example result of FACTS derived from the adult dataset [1]. The "if clause" represents the subgroup $G_p$, which contains all the affected individuals that satisfy the predicate $p = $ (*hours-per-week = FullTime*) $\wedge$ (*marital-status = Married-civ-spouse*) $\wedge$ (*occupation = Adm-clerical*). The information below the predicate refers to the protected subgroups $G_{p,0}$ and $G_{p,1}$ which are the female and male individuals of $G_p$ respectively. With blue color, we highlight the percentage $\mathsf{cov}(G_{p,i}) = |G_{p,i}|/|D_i|$ which serves as an indicator of the size of the protected subgroup. The most important part of the representation is the actions applied that appear below each protected subgroup and are evaluated in terms of fairness metrics. In this example, the metric is the *Equal Cost of Effectiveness* with effectiveness threshold $\phi = 0.7$. For $G_{p,0}$ there is no action surpassing the threshold $\phi = 0.7$, therefore we display a message accordingly. On the contrary, the action $a = \{$*hours-per-week* $\rightarrow$ *Overtime*, *occupation* $\rightarrow$ *Exec-managerial*$\}$ has effectiveness $0.72 > \phi$ for $G_{p,1}$, thus allowing a respective 72% of the male individuals of $G_p$ to achieve recourse. The unfairness score is "inf", since no recourse is achieved for the female subgroup.

```
If hours-per-week=FullTime, marital-status=Married-civ-spouse, occupation=Adm-clerical:
    Protected Subgroup = 'Male', 1.87% covered
        Make hours-per-week=Overtime, occupation=Exec-managerial with effectiveness 72.00%
    Protected Subgroup = 'Female', 1.80% covered
        No recourses for this subgroup.
    Bias against 'Female' due to Equal Cost of Effectiveness (threshold=0.7). Unfairness score = inf.
```

Figure 2: CSC for a highly biased subgroup in terms of *Equal Cost of Effectiveness* with $\phi = 0.7$.

**Method overview** Deploying FACTS comprises three main steps: (a) *Subgroup and action space generation* that creates the sets of groups $\mathcal{G}$ and actions $A$ to examine; (b) *Counterfactual summaries generation* that applies appropriate actions to each group $G \in \mathcal{G}$; and (c) *CSC construction and fairness ranking* that applies the definitions of Section 2.3. Next, we describe these steps in detail.

**(a) Subgroup and action space generation** Subgroups are generated by executing the fp-growth [11] frequent itemset mining algorithm on $D_0$ and on $D_1$ resulting to the sets of subgroups $\mathcal{G}_0$ and $\mathcal{G}_1$ and then by computing the intersection $\mathcal{G} = \mathcal{G}_0 \bigcap \mathcal{G}_1$. In our setting, an item is a feature-level predicate of the form "feature-operator-value" and, consequently, an itemset is a predicate $p$ defining a subgroup $G_p$. This step guarantees that the evaluation in terms of fairness will be performed

between the common subgroups $\mathcal{G}$ of the protected populations. The set of all actions $A$ is generated by executing fp-growth on the unaffected population to increase the chance of more effective actions and to reduce the computational complexity. The above process is parameterizable w.r.t. the selection of the protected attribute(s) and the minimum frequency threshold for obtaining candidate subgroups. We emphasize that alternate mechanisms to generate the sets $A$ and $\mathcal{G}$ are possible.

**(b) Counterfactual summaries generation** For each subgroup $G_p \in \mathcal{G}$, the following steps are performed: (i) Find valid actions, i.e., the actions in $A$ that contain a subset of the features appearing in $p$ and at least one different value in these features; (ii) For each valid action $a$ compute $\mathsf{eff}(a, G_{p,0})$ and $\mathsf{eff}(a, G_{p,1})$. The aforementioned process extracts, for each subgroup $G_p$, a subset $V_p$ of the actions $A$, with each action having exactly the same cost for all individuals of $G_p$. Therefore, individuals of $G_{p,0}$ and $G_{p,1}$ are evaluated in terms of subgroup-level actions, with a fixed cost for all individuals of the subgroup, in contrast to methods that rely on aggregating the cost of individual counterfactuals. This approach provides a key advantage to our method in cases where the definition of the exact costs for actions is either difficult or ambiguous: a misguided or even completely erroneous attribution of a cost to an action will equally affect all individuals of the subgroup and only to the extent that the respective fairness definition allows it. In the setting of individual counterfactual cost aggregation, changes in the same sets of features could lead to highly varying action costs for different individuals within the same subgroup.

**(c) CSC construction and fairness ranking** FACTS evaluates all definitions of Section 2.3 on all subgroups, producing an unfairness score per definition, per subgroup. In particular, each definition of Section 2.3 quantifies a different aspect of the difficulty for a protected subgroup to achieve recourse. This quantification directly translates to difficulty scores for the protected subgroups $G_{p,0}$ and $G_{p,1}$ of each subgroup $G_p$, which we compare accordingly (computing the absolute difference between them) to arrive at the unfairness score of each $G_p$ based on the particular fairness metric.

The outcome of this process is the generation, for each fairness definition, of a ranked list of CSC representations, in decreasing order of their unfairness score. Apart from unfairness ranking, the CSC representations will allow users to intuitively understand unfairness by directly comparing differences in actions between the protected populations within a subgroup.

## 4   Experiments

This section presents the experimental evaluation of FACTS on the Adult dataset [1]; more information about the datasets, experimental setting, and additional results can be found in the appendix. The code is available at: `https://github.com/AutoFairAthenaRC/FACTS`. First, we briefly describe the experimental setting and then present and discuss Comparative Subgroup Counterfactuals for subgroups ranked as the most unfair according to various definitions of Section 2.3.

Table 1: Unfair subgroups identified in the Adult dataset.

|  | Subgroup 1 | | | Subgroup 2 | | | Subgroup 3 | | |
|---|---|---|---|---|---|---|---|---|---|
|  | rank | bias against | unfairness score | rank | bias against | unfairness score | rank | bias against | unfairness score |
| Equal Effectiveness | 2950 | Male | 0.11 | 10063 | Female | 0.0004 | 275 | Female | 0.32 |
| Equal Choice for Recourse ($\phi = 0.3$) | Fair | - | 0 | 12 | Female | 2 | Fair | - | 0 |
| Equal Choice for Recourse ($\phi = 0.7$) | 6 | Male | 1 | **1** | Female | 6 | Fair | - | 0 |
| Equal Effectiveness within Budget ($c = 5$) | Fair | - | 0 | 2806 | Female | 0.056 | 70 | Female | 0.3 |
| Equal Effectiveness within Budget ($c = 10$) | 2350 | Male | 0.11 | 8518 | Female | 0.0004 | 226 | Female | 0.3 |
| Equal Effectiveness within Budget ($c = 18$) | 2675 | Male | 0.11 | 9222 | Female | 0.0004 | 272 | Female | 0.3 |
| Equal Cost of Effectiveness ($\phi = 0.3$) | Fair | - | 0 | Fair | - | 0 | **1** | Female | inf |
| Equal Cost of Effectiveness ($\phi = 0.7$) | **1** | Male | inf | 12 | Female | 2 | Fair | - | 0 |
| Fair Effectiveness-Cost Trade-Off | 4065 | Male | 0.11 | 3579 | Female | 0.13 | 306 | Female | 0.32 |
| Equal (Conditional) Mean Recourse | Fair | - | 0 | 3145 | Female | 0.35 | Fair | - | 0 |

**Experimental Setting** The first step was the dataset cleanup (e.g., removing missing values and duplicate features, creating bins for continuous features like age). The resulting dataset was split randomly with a 70:30 split ratio and was used to train and test respectively a logistic regression model (consequently used as the black-box model to audit). For the generation of the subgroups and the set of actions we used fp-growth with a 1% support threshold on the test set. We also implemented various cost functions, depending on the type of feature, i.e., categorical, ordinal, and numerical. A detailed description of the experimental setting, the models used, and the processes of our framework can be found in the supplementary material.

**Unfair subgroups** Table 1 presents three subgroups which were ranked at position 1 according to three different definitions: *Equal Cost of Effectiveness ($\phi = 0.7$)*, *Equal Choice for Recourse ($\phi = 0.7$)* and *Equal Cost of Effectiveness ($\phi = 0.3$)*, meaning that these subgroups were detected to have the highest unfairness according to the respective definitions. For each subgroup, its *rank*, *bias against*, and *unfairness score* are provided for all definitions presented in the left-most column. When the unfairness score is 0, we display the value "Fair" in the rank column. Note that subgroups with exactly the same score w.r.t. a definition will receive the same rank. The CSC representations for the fairness metric that ranked the three subgroups of Table 1 at the first position are shown in Figure 3.

Subgroup 1 is ranked first (highly unfair) based on *Equal Cost of Effectiveness with $\phi = 0.7$*, while it is ranked much lower or it is even considered as fair according to most of the remaining definitions (see the values of the "rank" column for Subgroup 1). The same pattern is observed for Subgroup 2 and Subgroup 3: they are ranked first based on *Equal Choice for Recourse with $\phi = 0.7$* and *Equal Cost of Effectiveness with $\phi = 0.3$* accordingly, but much lower according to the remaining definitions. This finding provides a strong indication of the utility of the different fairness definitions, i.e., the fact that they are able to capture different aspects of the difficulty in achieving recourse.[1]

```
Subgroup 1
If age=(41.0, 50.0], marital-status=Never-married, race=White, relationship=Not-in-family:
    Protected Subgroup = 'Male', 1.34% covered
        No recourses for this subgroup.
    Protected Subgroup = 'Female', 1.47% covered
        Make marital-status=Married-civ-spouse, relationship=Married with effectiveness 70.49%
    Bias against 'Male' due to Equal Cost of Effectiveness (threshold = 0.7). Unfairness score = inf.
```

```
Subgroup 2
If worklass=Private, hours-per-week=FullTime, marital-status=Married-civ-spouse, occupation=Adm-clerical, race=
        White:
  Protected Subgroup = 'Male', 1.04% covered
    Make hours-per-week=OverTime, occupation=Exec-managerial with effectiveness 70.00%
    Make hours-per-week=OverTime, occupation=Prof-specialty with effectiveness 70.00%
    Make hours-per-week=BrainDrain, occupation=Exec-managerial with effectiveness 70.00%
    Make hours-per-week=BrainDrain, occupation=Prof-specialty with effectiveness 70.00%
    Make Workclass=Self-emp-in, occupation=Exec-managerial with effectiveness 70.00%
    Make Workclass=Self-emp-in, hours-per-week=OverTime, occupation=Exec-managerial with effectiveness 80.00%
    Make Workclass=Self-emp-in, hours-per-week=OverTime, occupation=Sales with effectiveness 70.00%
    Make Workclass=Self-emp-in, hours-per-week=BrainDrain, occupation=Exec-managerial with effectiveness 70.00%
  Protected Subgroup = 'Female', 3.51% covered
    Make Workclass=Self-emp-in, hours-per-week=OverTime, occupation=Exec-managerial with effectiveness 74.51%
    Make Workclass=Self-emp-in, hours-per-week=BrainDrain, occupation=Exec-managerial with effectiveness 74.51%
  Bias against 'Female' due to Equal Choice for Recourse (threshold = 0.7). Unfairness score = 6
```

```
Subgroup 3
If age=(41.0, 50.0], occupation=Sales:
    Protected Subgroup 'Male', 1.18% covered
        Make occupation Craft-repair with effectiveness 0.00%
        Make occupation=Adm-clerical with effectiveness 0.00%
        Make occupation=Tech-support with effectiveness 19.23%
        Make occupation=Prof-specialty with effectiveness 28.21%
        Make occupation=Exec-managerial with effectiveness 39.74%
    Protected Subgroup 'Female', 1.56% covered
        Make occupation=Craft-repair with effectiveness 0.00%
        Make occupation=Adm-clerical with effectiveness 0.00%
        Make occupation=Tech-support with effectiveness 0.00%
        Make occupation=Exec-managerial with effectiveness 6.94%
        Make occupation=Prof-specialty with effectiveness 6.94%
        Make age=(50.0,90.0] with effectiveness 6.94%
        Make age=(50.0,90.0], occupation=Prof-specialty with effectiveness 6.94%
        Make age=(50.0,90.0], occupation=Craft-repair with effectiveness 6.94%
        Make age=(50.0,90.0], occupation=Adm-clerical with effectiveness 6.94%
        Make age=(50.0,90.0], occupation=Exec-managerial with effectiveness 6.94%
    Bias against 'Female' due to Equal Cost of Effectiveness (threshold = 0.3). Unfairness score = inf.
```

Figure 3: Comparative Subgroup Counterfactuals for the subgroups of Table 1.

What is more, these different aspects can easily be motivated by real-world auditing needs. For example, out of the aforementioned definitions, *Equal Cost of Effectiveness with $\phi = 0.7$* would be suitable in a scenario where a horizontal intervention to support a subpopulation needs to be performed, but a limited number of actions is affordable. In this case, the macro viewpoint demonstrated in the CSC Subgroup 1 (top result in Figure 3) serves exactly this purpose: one can easily derive single, group-level actions that can effectively achieve recourse for a desired percentage of the unfavored

---

[1]Additional examples, as well as statistics measuring this pattern on a significantly larger sample, are included in the supplementary material to further support this finding.

subpopulation. On the other hand, *Equal Choice for Recourse ($\phi = 0.3$)*, for which the CSC of a top result is shown in the middle of Figure 3, is mainly suitable for cases where assigning costs to actions might be cumbersome or even dangerous/unethical. This definition is oblivious to costs and measures bias based on the difference in the number of sufficiently effective actions to achieve recourse between the protected subgroups.

Another important observation comes again from Subgroup 1, where bias against the *Male* protected subgroup is detected, contrary to empirical findings from works employing more statistical-oriented approaches (e.g., [34]), where only subgroups with bias against the *Female* protected subgroup are reported. We deem this finding important, since it directly links to the problem of gerrymandering [17], which consists in partitioning attributes in such way and granularity to mask bias in subgroups. Our framework demonstrates robustness to this phenomenon, given that it can be properly configured to examine sufficiently small subgroups via the minimum frequency threshold described in Section 3.

## 5   Limitations

There exist numerous challenges and constraints when attempting to determine costs associated with specific counterfactuals. The cost functions involved are intricate, dataset-specific, and frequently necessitate the expertise of a domain specialist. These cost functions may depend on individual characteristics or human interpretation of the perceived difficulty of specific changes, potentially giving rise to concerns like breaches of user privacy (as the expert may require access to all individual user characteristics [25]) or the risk of malicious specialists manipulating cost functions to conceal existing biases.

We acknowledge the difficulties of finding the dataset-dependent cost functions and proceed to implement various "natural" cost functions tailored to different feature types (e.g., $L_1$ norm, exponential distances). These are only suggestions to the expert who is going to use our framework, are susceptible to change, and can be tailored for the specific datasets. Our primary focus is to identify potential biases towards specific subpopulations and recognizing the aforementioned inherent difficulties in defining costs, we introduce fairness metrics that remain independent of cost considerations. We recognize that a single metric, whether cost-dependent or not, cannot identify all forms of bias that may be present within a model's predictions. We do not aim to replace the existing statistical measures for bias detection, rather than complement them by focusing on the *bias of achieving recourse*.

It is essential to note that FACTS, like other resource generation algorithms, may be susceptible to errors, as demonstrated in prior research [31]. Outcomes produced by FACTS can vary based on the chosen configuration, including hyperparameters and metric selection. These variations can either obscure biases present within a classifier or indicate their absence. It is crucial to emphasize that since FACTS is an explainable framework for bias assessment within subgroups, its primary purpose is to serve as a guidance tool for auditors to identify subgroups requiring in-depth analysis of fairness criteria violations. Responsible usage of our framework necessitates transparency regarding the chosen configuration. Therefore, we recommend running FACTS with different hyperparameters and utilizing as many metrics as possible. Appendix G provides a comprehensive discussion of FACTS' limitations and responsible disclosure guidelines.

## 6   Conclusion

In this work, we delve deeper into the difficulty (or burden) of achieving recourse, an implicit and less studied type of bias. In particular, we go beyond existing works, by introducing a framework of fairness notions and definitions that, among others, conceptualizes the distinction between the micro and macro viewpoint of the problem, allows the consideration of a subgroup as a whole when exploring recourses, and supports *oblivious to action cost* fairness auditing. We complement this framework with an efficient implementation that allows the detection and ranking of subgroups according to the introduced fairness definitions and produces intuitive, explainable subgroup representations in the form of counterfactual summaries. Our next steps include the extension of the set of fairness definitions, focusing on the comparison of the effectiveness-cost distribution; improvements on the exploration, filtering, and ranking of the subgroup representations, via considering hierarchical relations or high-coverage of subgroups; a user study to evaluate the interpretability of fairness judgments, and the development of a fully-fledged auditing tool.

## Acknowledgments and Disclosure of Funding

This work has been funded by the European Union's Horizon Europe research and innovation programme under Grant Agreement No. 101070568 (AutoFair).

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

# Appendix

This is the appendix to the main paper, discussing how numerical attributes are handled (Appendix A), describing in detail the experimental setting (Appendix B), presenting the datasets (Appendix C), providing additional results (Appendix D) and discussion (Appndix E), quantitatively comparing the various fairness notions (Appendix F) and describing in detail limitations and usage guidelines of our framework to avoid malicious usage (Appendix G).

## A   Discussion about Numerical Attributes

FACTS works for datasets and models with numerical or continuous features; in fact it works with any mixture of categorical and numerical features. The core idea is that numerical attributes are binned. These bins are used to define subpopulations, e.g., people in the salary range $[40K, 50K]$, and actions, e.g., make salary $[50K, 60K]$. So the action "if salary is $[40K, 50K]$, make salary $[50K, 60K]$" means that all individuals within that salary range are mapped to their counterfactuals where their salaries are increased by $10K$.

Binning is necessary when defining subpopulations, as we want to explore the entire feature space and present conclusions in a manner that is easily understandable by humans, e.g., "there is unfairness against females when married=no, salary in $[40K, 50K]$".

Binning is also necessary when considering actions over numerical attributes. As explained, binning of granularity $10K$ for salary means that we consider actions that change salary by $\pm 10K$, $\pm 20K$, etc. We emphasize that because the number of actions/counterfactuals is infinite, all methods that work with model-agnostic counterfactual-based explanations (e.g., FACTS, and [31, 25]) have to explore only a small set of actions.

Our goal is to approximate the effectiveness-cost distribution, which means we should consider as many actions as possible. Therefore, on the one hand, binning for actions should rather be fine-grained, while on the other hand, binning for subpopulations should be moderately coarse-grained so as to draw conclusions that concern many affected individuals.

With that said, the binning granularity for actions and subpopulations can differ. For example consider bins of length $10K$ for subpopulations and $5K$ for actions. An action, "if salary is $[40K, 50K]$, make salary $[55K, 60K]$" means that individuals with salary in $[40K, 45K]$ increase their salary by $15K$, and individuals with salary in $[45K, 50K]$ increase their salary by $10K$.

# B  Experimental Setting

**Models**   To conduct our experiments, we have used the Logistic Regression[2] classification model, where we use the default implementation of the python package `scikit-learn`[3]. This model corresponds to the black box one that our framework audits in terms of fairness of recourse.

**Train-Test Split**   For our experiments, all datasets are split into training and test sets with proportions 70% and 30%, respectively. Both shuffling of the data and stratification based on the labels were employed. Our results can be reproduced using the random seed value 131313 in the data split function (`train_test_split`[4] from the python package `scikit-learn`). FACTS is deployed solely on the test set.

**Frequent Itemset Mining**   The set of subgroups and the set of actions are generated by executing the fp-growth[5] algorithm for frequent itemset mining. We used the implementation in the Python package `mlxtend` [6]. We deploy fp-growth with support threshold **1%**, i.e., we require the return of subgroups and actions with at least 1% frequency in the respective populations. Recall that subgroups are derived from the affected populations $D_0$ and $D_1$ and actions are derived from the unaffected population.

**Effectiveness and Budgets**   As we have stated in Section 2 of our main paper, the metrics *Equal Choice for Recourse* and *Equal Cost of Effectiveness* require the definition of a target effectiveness level $\phi$, while the metric *Equal Effectiveness within Budget* requires the definition of a target cost level (or budget) $c$.

Regarding the metrics that require the definition of an effectiveness level $\phi$, we used two different values arbitrarily, i.e., a relatively low effectiveness level of $\phi = 30\%$ and a relatively high effectiveness level of $\phi = 70\%$.

For the estimation of budget-level values $c$ we followed a more elaborate procedure. Specifically,

1. Compute the *Equal Cost of Effectiveness* (micro definition) with a target effectiveness level of $\phi = 50\%$ to calculate, for all subgroups $G$, the minimum cost required to flip the prediction for at least 50% of both $G_0$ and $G_1$.
2. Gather all such minimum costs of step 1 in an array.
3. Choose budget values as percentiles of this set of cost values. We have chosen the **30%, 60%** and **90%** percentiles arbitrarily.

**Cost Functions**   Our implementation allows the user to define any cost function based on their domain knowledge and requirements. For evaluation and demonstration purposes, we implement an indicative set of cost functions, according to which, the cost of a change of a feature value $v$ to the value $v'$ is defined as follows:

1. **Numerical features:** $|norm(v) - norm(v')|$, where $norm$ is a function that normalizes values to $[0, 1]$.
2. **Categorical features:** 1 if $v \neq v'$, and 0 otherwise.
3. **Ordinal features:** $|pos(v) - pos(v')|$, where $pos$ is a function that provides the order for each value.

Additionally to the above costs, the user is able to define a feature-specific weight that indicates the difficulty of changing the given feature through an action. Thus, for each dataset, the cost of actions can be simply determined by specifying the numerical, categorical, and ordinal features, as well as the weights for each feature.

---

[2]`https://scikit-learn.org/stable/modules/generated/sklearn.linear_model.`
`LogisticRegression.html`
[3]`https://scikit-learn.org/stable/index.html`
[4]`https://scikit-learn.org/stable/modules/generated/sklearn.model_selection.train_`
`test_split.html`
[5]`https://rasbt.github.io/mlxtend/user_guide/frequent_patterns/fpgrowth/`
[6]`https://github.com/rasbt/mlxtend`

**Feasibility**   Apart from the cost of actions, we also take care of some obvious unfeasible actions such as that the age and education features can not be reduced and actions should not lead to unknown or missing values.

**Compute resources**   Experiments were run on commodity hardware (AMD Ryzen 5 5600H processor, 8GB RAM). On the software side, all experiments were run in an isolated conda environment using Python 3.9.16.

# C  Datasets Description

We have used five datasets in our experimental evaluation; the main paper presented results only on the first. For each dataset, we provide details about the preprocessing procedure, specify feature types, and list the cost feature weights applied.

## C.1  Adult

We have generated CSCs in the Adult dataset[7] using two different features as protected attributes, i.e., 'sex', and 'race'. The assessment of bias for each protected attribute is done separately. The results for 'sex' as the protected attribute are presented in the main paper. Before we present our results for race as the protected attribute, we briefly discuss the preprocessing procedures and feature weights used for the adult dataset.

**Preprocessing**  We removed the features 'fnlwgt' and 'education' and any rows with unknown values. The 'hours-per-week' and 'age' features have been discretized into 5 bins each.

**Features**  All features have been treated as categorical, except for 'capital-gain' and 'capital-loss', which are numeric, and 'education-num' and 'hours-per-week', which we treat as ordinal. The feature weights that we used for the cost function are presented in Table 2. We need to remind here that this comprises only an indicative weight assignment to serve our experimentation; the weight below try to capture the notion of how feasible/actionable it is to perform a change to a specific feature.

Table 2: Cost Feature Weights for Adult

| feature name | weight value | feature name | weight value |
|---|---|---|---|
| native-country | 4 | Workclass | 2 |
| marital-status | 5 | hours-per-week | 2 |
| relationship | 5 | capital-gain | 1 |
| age | 10 | capital-loss | 1 |
| occupation | 4 | education-num | 3 |

## C.2  COMPAS

We have generated CSCs in the COMPAS dataset[8] for race as the protected attribute. Apart from our results, we provide some brief information regarding preprocessing procedures and the cost feature weights for the COMPAS dataset.

**Preprocessing**  We discard the features 'age' and 'c_charge_desc'. The 'priors_count' feature has been discretized into 5 bins: [-0.1,1), [1, 5) [5, 10) [10, 15) and [15, 38), while trying to keep the frequencies of each bin approximately equal (the distribution of values is highly asymmetric so this is not possible with the direct use of e.g., `pandas.qcut`[9]).

**Features**  We treat the features 'juv_fel_count', 'juv_misd_count', 'juv_other_count' as numerical and the rest as categorical. The feature weights used for the cost function are shown in Table 3.

## C.3  SSL

We have generated CSCs in the SSL dataset[10] for race as the protected attribute. Before we move to our results, we discuss briefly preprocessing procedures and feature weights applied in the SSL dataset.

---

[7] `https://raw.githubusercontent.com/columbia/fairtest/master/data/adult/adult.csv`
[8] `https://aif360.readthedocs.io/en/latest/modules/generated/aif360.sklearn.` `datasets.fetch_compas.html`
[9] `https://pandas.pydata.org/pandas-docs/stable/reference/api/pandas.qcut.html`
[10] `https://raw.githubusercontent.com/samuel-yeom/fliptest/master/exact-ot/` `chicago-ssl-clean.csv`

Table 3: Cost Feature Weights for COMPAS

| feature name | weight value |
|---|---|
| age_cat | 10 |
| juv_fel_count | 1 |
| juv_fel_count | 1 |
| juv_other_count | 1 |
| priors_count | 1 |
| c_charge_degree | 1 |

**Preprocessing**    We remove all rows with missing values ('U' or 'X') from the dataset. We also discretize the feature 'PREDICTOR RAT TREND IN CRIMINAL ACTIVITY' into 6 bins. Finally, since the target labels are values between 0 and 500, we 'binarize' them by assuming values above 344 to be positively impacted and below 345 negatively impacted (following the principles used in [3].

**Features**    In this dataset, we treat all features as numerical (apart from the protected race feature). Table 4 presents the feature weights used for the cost function.

Table 4: Cost Feature Weights for SSL

| feature name | weight value |
|---|---|
| PREDICTOR RAT AGE AT LATEST ARREST | 10 |
| PREDICTOR RAT VICTIM SHOOTING INCIDENTS | 1 |
| PREDICTOR RAT VICTIM BATTERY OR ASSAULT | 1 |
| PREDICTOR RAT ARRESTS VIOLENT OFFENSES | 1 |
| PREDICTOR RAT GANG AFFILIATION | 1 |
| PREDICTOR RAT NARCOTIC ARRESTS | 1 |
| PREDICTOR RAT TREND IN CRIMINAL ACTIVITY | 1 |
| PREDICTOR RAT UUW ARRESTS | 1 |

## C.4   Ad Campaign

We have generated CSCs in the Ad Campaign dataset[11] for gender as the protected attribute.

**Preprocessing**    We decided not to remove missing values, since they represent the vast majority of values for all features. However, we did not allow actions that lead to missing values in the CSCs representation.

Table 5: The assigned weights for each feature used in the calculation of the cost function in the Ad Campaign dataset

| feature name | weight value |
|---|---|
| religion | 5 |
| politics | 2 |
| parents | 3 |
| age | 10 |
| income | 3 |
| area | 2 |
| college_educated | 3 |
| homeowner | 1 |

---

[11]https://developer.ibm.com/exchanges/data/all/bias-in-advertising/

**Features**  In this dataset, we treat all features, apart from the protected one, as categorical. The feature weights used for the cost function are shown in Table 5.

## C.5  Credit

We have generated CSCs in the Statlog (German Credit) dataset[12] for sex as the protected attribute.

**Preprocessing**  We performed only minimal preprocessing. This involved adding column names, based on the provided documentation, since they are not automatically included in the dataset. We also simplified the values of the 'sex' feature into 'Male' and 'Female', from their original, more complex form (which originally was included in a coded version of the combination of personal status and sex). Finally, we changed the labels from {1, 2} to {0, 1}.

**Features**  The features 'duration', 'credit', 'rate', 'residence', 'age', 'cards', and 'liables' are treated as numerical, while the remaining features are regarded as categorical. We assigned the same weight value, specifically 1, for all features in the dataset when calculating the cost function, as depicted in Table 6.

Table 6: The assigned weights for each feature used in the calculation of the cost function in the German Credit dataset

| feature name | weight value |
|---|---|
| status | 1 |
| duration | 1 |
| hist | 1 |
| purpose | 1 |
| credit | 1 |
| savings | 1 |
| empl | 1 |
| rate | 1 |
| sex | 1 |
| others | 1 |
| residence | 1 |
| property | 1 |
| age | 1 |
| plans | 1 |
| housing | 1 |
| cards | 1 |
| job | 1 |
| liables | 1 |
| telephone | 1 |
| foreign | 1 |

---

[12]https://archive.ics.uci.edu/dataset/144/statlog+german+credit+data

# D  Additional Results

This section repeats the experiment described in the main paper, concerning the Adult dataset with 'gender' as the protected attribute (Section 4), in three other cases. Specifically, we provide three subgroups that were ranked first in terms of unfairness according to a metric, highlight why they were marked as unfair by our framework, and summarize their unfairness scores according to the rest of the metrics.

## D.1  Results for Adult with race as the protected attribute

We showcase three prevalent subgroups for which the rankings assigned by different fairness definitions truly yield different kinds of information. This is showcased in Table 7. We once again note that the results presented here are for 'race' as a protected attribute, while the corresponding results for 'gender' are presented in Section 4 of the main paper.

Table 7: Example of three unfair subgroups in Adult (protected attribute race)

| | Subgroup 1 | | | Subgroup 2 | | | Subgroup 3 | | |
|---|---|---|---|---|---|---|---|---|---|
| | rank | bias against | unfairness score | rank | bias against | unfairness score | rank | bias against | unfairness score |
| Equal Effectiveness | Fair | Fair | 0.0 | 3047.0 | Non-White | 0.115 | 1682.0 | Non-White | 0.162 |
| Equal Choice for Recourse ($\phi = 0.3$) | **1** | Non-White | 10.0 | 10.0 | Non-White | 1.0 | Fair | Fair | 0.0 |
| Equal Choice for Recourse ($\phi = 0.7$) | Fair | Fair | 0.0 | Fair | Fair | 0.0 | Fair | Fair | 0.0 |
| Equal Effectiveness within Budget (c = 1.15) | Fair | Fair | 0.0 | Fair | Fair | 0.0 | Fair | Fair | 0.0 |
| Equal Effectiveness within Budget (c = 10.0) | 303.0 | Non-White | 0.242 | 2201.0 | Non-White | 0.115 | 4035.0 | Non-White | 0.071 |
| Equal Effectiveness within Budget (c = 21.0) | Fair | Fair | 0.0 | 2978.0 | Non-White | 0.115 | 1663.0 | Non-White | 0.162 |
| Equal Cost of Effectiveness ($\phi = 0.3$) | 18.0 | Non-White | 0.15 | **1** | Non-White | inf | Fair | Fair | 0.0 |
| Equal Cost of Effectiveness ($\phi = 0.7$) | Fair | Fair | 0.0 | Fair | Fair | 0.0 | Fair | Fair | 0.0 |
| Fair Effectiveness-Cost Trade-Off | 909.0 | Non-White | 0.242 | 4597.0 | Non-White | 0.115 | 2644.0 | Non-White | 0.162 |
| Equal (Conditional) Mean Recourse | 5897.0 | White | 0.021 | 5309.0 | White | 0.047 | **1** | Non-White | inf |

In Figure 4 we present the Comparative Subgroup Counterfactual representation for the subgroups of Table 7 that corresponds to the fairness metric for which each subgroup presents the minimum rank.

These results are in line with the findings reported in the main paper (Section 4), on the same dataset (Adult), but on a different protected attribute (race instead of gender). Subgroups ranked first (highly unfair) with respect to a specific definition, are ranked much lower or even considered as fair according to most of the remaining definitions. This serves as an indication of the utility of the different fairness definitions, which is further strengthened by the diversity of the respective CSCs of Table 7. For example, the Subgroup 1 CSC (ranked first *Equal Choice for Recourse ($\phi = 0.3$)*), demonstrates unfairness by contradicting a plethora of actions for the "White" protected subgroup, as opposed to much fewer actions for the the "Non-White" protected subgroup. For Subgroup 2, a much more concise representation is provided, tied to the respective definition (*Equal Cost of Effectiveness ($\phi = 0.3$)*): no recourses are identified for the desired percentage of the "Non-White" unfavored population, as opposed to the "White" unfavored population.

## D.2  Results for COMPAS

We present some ranking statistics for three interesting subgroups for all fairness definitions (Table 8). The Comparative Subgroup Counterfactuals for the same three subgroups are shown in Figure 5.

Table 8: Example of three unfair subgroups in COMPAS

| | Subgroup 1 | | | Subgroup 2 | | | Subgroup 3 | | |
|---|---|---|---|---|---|---|---|---|---|
| | rank | bias against | unfairness score | rank | bias against | unfairness score | rank | bias against | unfairness score |
| Equal Effectiveness | Fair | Fair | 0.0 | 116.0 | African-American | 0.151 | 209.0 | African-American | 0.071 |
| Equal Choice for Recourse ($\phi = 0.3$) | Fair | Fair | 0.0 | 3.0 | African-American | 1.0 | Fair | Fair | 0.0 |
| Equal Choice for Recourse ($\phi = 0.7$) | **1** | African-American | 3.0 | Fair | Fair | 0.0 | Fair | Fair | 0.0 |
| Equal Effectiveness within Budget (c = 1) | 66.0 | African-American | 0.167 | 79.0 | African-American | 0.151 | 185.0 | African-American | 0.071 |
| Equal Effectiveness within Budget (c = 10) | 84.0 | African-American | 0.167 | 108.0 | African-American | 0.151 | 220.0 | African-American | 0.071 |
| Equal Cost of Effectiveness ($\phi = 0.3$) | Fair | Fair | 0.0 | **1** | African-American | inf | Fair | Fair | 0.0 |
| Equal Cost of Effectiveness ($\phi = 0.7$) | Fair | Fair | 0.0 | Fair | Fair | 0.0 | Fair | Fair | 0.0 |
| Fair Effectiveness-Cost Trade-Off | 3.0 | African-American | 0.5 | 214.0 | African-American | 0.151 | 376.0 | African-American | 0.071 |
| Equal (Conditional) Mean Recourse | 59.0 | African-American | 1.667 | Fair | Fair | 0.0 | **1** | African-American | inf |

## D.3  Results for SSL

In Table 9 we present a summary of the ranking statistics for three interesting subgroups. and their respective Comparative Subgroup Counterfactuals in Figure 6.

```
Subgroup 1
If workclass=Private , age=(34.0 , 41.0] , capital−gain=0 , capital−loss=0 , marital−status=Never−married , native−
    country=United−States , relationship=Not−in−family :
    Protected Subgroup = 'Non−White' , 1.09% covered
        Make Workclas=Federal−gov , age=(41.0 , 50.0] , marital−status=Married−civ−spouse , relationship=Married
                with effectiveness 36.84%.
        Make capital−gain=15024 , marital−status=Married−civ−spouse , relationship=Married with effectiveness
                100.00%.
        Make age=(41.0 , 50.0] , capital−gain=15024 , marital−status=Married−civ−spouse , relationship=Married
                with effectiveness 100.00%.
    Protected Subgroup = 'White' , 1.94% covered
        Make age=(41.0 , 50.0] , marital−status=Married−civ−spouse , relationship=Married with effectiveness
                45.14%.
        Make marital−status=Married−civ−spouse , relationship=Married with effectiveness 40.00%.
        Make age=(50.0 , 90.0] , marital−status= Married−civ−spouse , relationship=Married with effectiveness
                42.86%.
        Make Workclass=Local−gov , age=(41.0 , 50.0] , marital−status=Married−civ−spouse , relationship=Married
                with effectiveness 44.00%.
        Make Workclass=Local−gov , age=(41.0 , 50.0] , marital−status=Married−civ−spouse , relationship=Married
                with effectiveness 44.00%.
        Make Workclas=Self−emp−inc , age=(41.0 , 50.0] , marital−status=Married−civ−spouse , relationship=Married
                with effectiveness 52.57%.
        Make Workclass=Self−emp−inc , age=(50.0 , 90.0] , marital−status=Married−civ−spouse , relationship=Married
                 with effectiveness 49.71%.
        Make Workclass=Local−gov , marital−status=Married−civ−spouse , relationship=Married with effectiveness
                36.00%.
        Make Workclass=Self−emp−inc , marital−status=Married−civ−spouse , relationship=Married with
                effectiveness 45.14%.
        Make Workclass= Federal−gov , age=(41.0 , 50.0] , marital−status=Married−civ−spouse , relationship=Married
                 with effectiveness 62.29%.
        Make Workclass=State−gov , age=(41.0 , 50.0] , marital−status=Married−civ−spouse , relationship=Married
                with effectiveness 40.57%.
        Make capital−gain=15024 , marital−status=Married−civ−spouse , relationship=Married with effectiveness
                99.43%.
        Make Workclass=Local−gov , age=(50.0 , 90.0] , marital−status=Married−civ−spouse , relationship=Married
                with effectiveness 40.00%.
        Make age=(41.0 , 50.0] , capital−gain=15024 , marital−status=Married−civ−spouse , relationship=Married
                with effectiveness 100.00%.
    Bias against 'Non−White' due to Equal Choice for Recourse (threshold = 0.3). Unfairness score = 10.
```

```
Subgroup 2
If hours−per−week = FullTime , native−country = United−States , occupation = Adm−clerical , relationship =
    Married :
    Protected Subgroup = 'Non−White' , 1.66% covered
        No recourses for this subgroup.
    Protected Subgroup = 'White' , 1.66% covered
        Make hours−per−week = BrainDrain , occupation = Exec−managerial with effectiveness 70.00%.
    Bias against 'Non−White' due to Equal Cost of Effectiveness (threshold = 0.3). Unfairness score = inf.
```

```
Subgroup 3
If hours−per−week = PartTime , marital−status = Divorced , native−country = United−States :
    Protected Subgroup = 'Non−White' , 1.15% covered
        Make marital−status=Married−civ−spouse with effectiveness 0.00%.
        Make hours−per−week=MidTime , marital−statu=Married−civ−spouse with effectiveness 0.00%.
        Make hours−per−week=FullTime , marital−status=Married−civ−spouse with effectiveness 0.00%.
        Make hours−per−week=OverTime , marital−status=Married−civ−spouse with effectiveness 0.00%.
        Make hours−per−week=OverTime , marital−status=Never−married with effectiveness 0.00%.
        Make hours−per−week=BrainDrain , marital−status=Married−civ−spouse with effectiveness 0.00%.
    Protected Subgroup = 'White' , 1.66% covered
        Make marital−status=Married−civ−spouse with effectiveness 1.01%.
        Make hours−per−week=MidTime , marital−statu=Married−civ−spouse with effectiveness 1.01%.
        Make hours−per−week=FullTime , marital−status=Married−civ−spouse with effectiveness 7.07%.
        Make hours−per−week=OverTime , marital−status=Married−civ−spouse with effectiveness 7.07%.
        Make hours−per−week=OverTime , marital−status=Never−married with effectiveness 15.15%.
        Make hours−per−week=BrainDrain , marital−status=Married−civ−spouse with effectiveness 16.16%.
    Bias against 'Non−White' due to Equal Conditional Mean Recourse. Unfairness score = inf.
```

Figure 4: Example of three Comparative Subgroup Counterfactuals in Adult (protected attribute race); ref. Table 7

## D.4   Results for Ad Campaign

In Table 10 we present, as we did for the other datasets, the ranking results for 3 interesting subgroups, while in Figure 7, we show the respective Comparative Subgroup Counterfactuals for these subgroups.

Figure 5: Example of three Comparative Subgroup Counterfactuals in COMPAS; ref. Table 8

Table 9: Example of three unfair subgroups in SSL

| | Subgroup 1 | | | Subgroup 2 | | | Subgroup 3 | | |
|---|---|---|---|---|---|---|---|---|---|
| | rank | bias against | unfairness score | rank | bias against | unfairness score | rank | bias against | unfairness score |
| Equal Effectiveness | 1630.0 | Black | 0.076 | 70.0 | Black | 0.663 | 979.0 | Black | 0.151 |
| Equal Choice for Recourse ($\phi = 0.3$) | Fair | Fair | 0.0 | 12.0 | Black | 1.0 | 12.0 | Black | 1.0 |
| Equal Choice for Recourse ($\phi = 0.7$) | 13.0 | Black | 3.0 | Fair | Fair | 0.0 | Fair | Fair | 0.0 |
| Equal Effectiveness within Budget (c = 1) | Fair | Fair | 0.0 | 195.0 | Black | 0.663 | 1692.0 | White | 0.138 |
| Equal Effectiveness within Budget (c = 2) | 2427.0 | Black | 0.111 | 126.0 | Black | 0.663 | 3686.0 | White | 0.043 |
| Equal Effectiveness within Budget (c = 10) | 2557.0 | Black | 0.076 | 73.0 | Black | 0.663 | 1496.0 | Black | 0.151 |
| Equal Cost of Effectiveness ($\phi = 0.3$) | Fair | Fair | 0.0 | 1 | Black | inf | 1 | Black | inf |
| Equal Cost of Effectiveness ($\phi = 0.7$) | 1 | Black | inf | Fair | Fair | 0.0 | Fair | Fair | 0.0 |
| Fair Effectiveness-Cost Trade-Off | 3393.0 | Black | 0.111 | 443.0 | Black | 0.663 | 2685.0 | Black | 0.151 |
| Equal (Conditional) Mean Recourse | 3486.0 | Black | 0.053 | 1 | Black | inf | 1374.0 | White | 0.95 |

Table 10: Example of three unfair subgroups in Ad Campaign

| | Subgroup 1 | | | Subgroup 2 | | | Subgroup 3 | | |
|---|---|---|---|---|---|---|---|---|---|
| | rank | bias against | unfairness score | rank | bias against | unfairness score | rank | bias against | unfairness score |
| Equal Effectiveness | 319.0 | Female | 0.286 | Fair | Fair | 0.0 | 467.0 | Male | 0.099 |
| Equal Choice for Recourse ($\phi = 0.3$) | 5.0 | Female | 1.0 | 2.0 | Female | 4.0 | Fair | Fair | 0.0 |
| Equal Choice for Recourse ($\phi = 0.7$) | Fair | Fair | 0.0 | 1 | Female | 4.0 | Fair | Fair | 0.0 |
| Equal Effectiveness within Budget (c = 1) | Fair | Fair | 0.0 | Fair | Fair | 0.0 | Fair | Fair | 0.0 |
| Equal Effectiveness within Budget (c = 5) | Fair | Fair | 0.0 | Fair | Fair | 0.0 | 345.0 | Male | 0.099 |
| Equal Cost of Effectiveness ($\phi = 0.3$) | 1 | Female | inf | Fair | Fair | 0.0 | Fair | Fair | 0.0 |
| Equal Cost of Effectiveness ($\phi = 0.7$) | Fair | Fair | 0.0 | Fair | Fair | 0.0 | Fair | Fair | 0.0 |
| Fair Effectiveness-Cost Trade-Off | 331.0 | Female | 0.286 | Fair | Male | 0.0 | 547.0 | Male | 0.099 |
| Equal (Conditional) Mean Recourse | Fair | Fair | 0.0 | Fair | Fair | 0.0 | 1 | Male | inf |

**Subgroup 1**

If PREDICTOR RAT ARRESTS VIOLENT OFFENSES = 1, PREDICTOR RAT NARCOTIC ARRESTS = 1, PREDICTOR RAT VICTIM BATTERY OR ASSAULT = 1:

Protected Subgroup = 'Black', 1.04% covered
  No recourses for this subgroup.
Protected Subgroup = 'White', 1.00% covered
  Make PREDICTOR RAT ARRESTS VIOLENT OFFENSES = 0, PREDICTOR RAT NARCOTIC ARRESTS = 0, PREDICTOR RAT VICTIM BATTERY OR ASSAULT = 0 with effectiveness 72.73%
  Make PREDICTOR RAT ARRESTS VIOLENT OFFENSES = 0, PREDICTOR RAT NARCOTIC ARRESTS = 1, PREDICTOR RAT VICTIM BATTERY OR ASSAULT = 0 with effectiveness 72.73%
  Make PREDICTOR RAT ARRESTS VIOLENT OFFENSES = 0, PREDICTOR RAT NARCOTIC ARRESTS = 2, PREDICTOR RAT VICTIM BATTERY OR ASSAULT = 0 with effectiveness 72.73%
Bias against 'Black' due to Equal Cost of Effectiveness (threshold = 0.7). Unfairness score = inf.

---

**Subgroup 2**

If PREDICTOR RAT NARCOTIC ARRESTS = 0, PREDICTOR RAT TREND IN CRIMINAL ACTIVITY = (−0.2, −0.1], PREDICTOR RAT UUW ARRESTS = 0, PREDICTOR RAT VICTIM SHOOTING INCIDENTS = 0:

Protected Subgroup = 'Black', 2.51% covered
  Make PREDICTOR RAT NARCOTIC ARRESTS = 0, PREDICTOR RAT TREND IN CRIMINAL ACTIVITY = (−8.200999999999999, −0.3], PREDICTOR RAT UUW ARRESTS = 0, PREDICTOR RAT VICTIM SHOOTING INCIDENTS = 0 with effectiveness 0.00%
  Make PREDICTOR RAT NARCOTIC ARRESTS = 0, PREDICTOR RAT TREND IN CRIMINAL ACTIVITY = (−0.1, 0.1], PREDICTOR RAT UUW ARRESTS = 0, PREDICTOR RAT VICTIM SHOOTING INCIDENTS = 0 with effectiveness 0.00%
  Make PREDICTOR RAT NARCOTIC ARRESTS = 0, PREDICTOR RAT TREND IN CRIMINAL ACTIVITY = (0.1, 0.3], PREDICTOR RAT UUW ARRESTS = 0, PREDICTOR RAT VICTIM SHOOTING INCIDENTS = 0 with effectiveness 0.00%
  Make PREDICTOR RAT NARCOTIC ARRESTS = 1, PREDICTOR RAT TREND IN CRIMINAL ACTIVITY = (−8.200999999999999, −0.3], PREDICTOR RAT UUW ARRESTS = 0, PREDICTOR RAT VICTIM SHOOTING INCIDENTS = 0 with effectiveness 0.00%
  Make PREDICTOR RAT NARCOTIC ARRESTS = 0, PREDICTOR RAT TREND IN CRIMINAL ACTIVITY = (0.3, 7.3], PREDICTOR RAT UUW ARRESTS = 0, PREDICTOR RAT VICTIM SHOOTING INCIDENTS = 0 with effectiveness 0.00%
  Make PREDICTOR RAT NARCOTIC ARRESTS = 0, PREDICTOR RAT TREND IN CRIMINAL ACTIVITY = (−0.3, −0.2], PREDICTOR RAT UUW ARRESTS = 0, PREDICTOR RAT VICTIM SHOOTING INCIDENTS = 0 with effectiveness 0.00%
  Make PREDICTOR RAT NARCOTIC ARRESTS = 1, PREDICTOR RAT TREND IN CRIMINAL ACTIVITY = (−0.1, 0.1], PREDICTOR RAT UUW ARRESTS = 0, PREDICTOR RAT VICTIM SHOOTING INCIDENTS = 0 with effectiveness 0.00%
  Make PREDICTOR RAT NARCOTIC ARRESTS = 1, PREDICTOR RAT TREND IN CRIMINAL ACTIVITY = (−0.2, −0.1], PREDICTOR RAT UUW ARRESTS = 0, PREDICTOR RAT VICTIM SHOOTING INCIDENTS = 0 with effectiveness 0.00%
  Make PREDICTOR RAT NARCOTIC ARRESTS = 1, PREDICTOR RAT TREND IN CRIMINAL ACTIVITY = (0.1, 0.3], PREDICTOR RAT UUW ARRESTS = 0, PREDICTOR RAT VICTIM SHOOTING INCIDENTS = 0 with effectiveness 0.00%
  Make PREDICTOR RAT NARCOTIC ARRESTS = 2, PREDICTOR RAT TREND IN CRIMINAL ACTIVITY = (−8.200999999999999, −0.3], PREDICTOR RAT UUW ARRESTS = 0, PREDICTOR RAT VICTIM SHOOTING INCIDENTS = 0 with effectiveness 0.00%
  Make PREDICTOR RAT NARCOTIC ARRESTS = 1, PREDICTOR RAT TREND IN CRIMINAL ACTIVITY = (0.3, 7.3], PREDICTOR RAT UUW ARRESTS = 0, PREDICTOR RAT VICTIM SHOOTING INCIDENTS = 0 with effectiveness 0.00%
Protected Subgroup = 'White', 2.87% covered
  Make PREDICTOR RAT NARCOTIC ARRESTS = 0, PREDICTOR RAT TREND IN CRIMINAL ACTIVITY = (−8.200999999999999, −0.3], PREDICTOR RAT UUW ARRESTS = 0, PREDICTOR RAT VICTIM SHOOTING INCIDENTS = 0 with effectiveness 57.14%
  Make PREDICTOR RAT NARCOTIC ARRESTS = 0, PREDICTOR RAT TREND IN CRIMINAL ACTIVITY = (−0.1, 0.1], PREDICTOR RAT UUW ARRESTS = 0, PREDICTOR RAT VICTIM SHOOTING INCIDENTS = 0 with effectiveness 0.00%
  Make PREDICTOR RAT NARCOTIC ARRESTS = 0, PREDICTOR RAT TREND IN CRIMINAL ACTIVITY = (0.1, 0.3], PREDICTOR RAT UUW ARRESTS = 0, PREDICTOR RAT VICTIM SHOOTING INCIDENTS = 0 with effectiveness 0.00%
  Make PREDICTOR RAT NARCOTIC ARRESTS = 1, PREDICTOR RAT TREND IN CRIMINAL ACTIVITY = (−8.200999999999999, −0.3], PREDICTOR RAT UUW ARRESTS = 0, PREDICTOR RAT VICTIM SHOOTING INCIDENTS = 0 with effectiveness 0.00%
  Make PREDICTOR RAT NARCOTIC ARRESTS = 0, PREDICTOR RAT TREND IN CRIMINAL ACTIVITY = (0.3, 7.3], PREDICTOR RAT UUW ARRESTS = 0, PREDICTOR RAT VICTIM SHOOTING INCIDENTS = 0 with effectiveness 0.00%
  Make PREDICTOR RAT NARCOTIC ARRESTS = 0, PREDICTOR RAT TREND IN CRIMINAL ACTIVITY = (−0.3, −0.2], PREDICTOR RAT UUW ARRESTS = 0, PREDICTOR RAT VICTIM SHOOTING INCIDENTS = 0 with effectiveness 0.00%
  Make PREDICTOR RAT NARCOTIC ARRESTS = 1, PREDICTOR RAT TREND IN CRIMINAL ACTIVITY = (−0.1, 0.1], PREDICTOR RAT UUW ARRESTS = 0, PREDICTOR RAT VICTIM SHOOTING INCIDENTS = 0 with effectiveness 0.00%
  Make PREDICTOR RAT NARCOTIC ARRESTS = 1, PREDICTOR RAT TREND IN CRIMINAL ACTIVITY = (−0.2, −0.1], PREDICTOR RAT UUW ARRESTS = 0, PREDICTOR RAT VICTIM SHOOTING INCIDENTS = 0 with effectiveness 0.00%
  Make PREDICTOR RAT NARCOTIC ARRESTS = 1, PREDICTOR RAT TREND IN CRIMINAL ACTIVITY = (0.1, 0.3], PREDICTOR RAT UUW ARRESTS = 0, PREDICTOR RAT VICTIM SHOOTING INCIDENTS = 0 with effectiveness 0.00%
  Make PREDICTOR RAT NARCOTIC ARRESTS = 2, PREDICTOR RAT TREND IN CRIMINAL ACTIVITY = (−8.200999999999999, −0.3], PREDICTOR RAT UUW ARRESTS = 0, PREDICTOR RAT VICTIM SHOOTING INCIDENTS = 0 with effectiveness 0.00%
  Make PREDICTOR RAT NARCOTIC ARRESTS = 1, PREDICTOR RAT TREND IN CRIMINAL ACTIVITY = (0.3, 7.3], PREDICTOR RAT UUW ARRESTS = 0, PREDICTOR RAT VICTIM SHOOTING INCIDENTS = 0 with effectiveness 0.00%
Bias against 'Black' due to Equal(Conditional) Mean Recourse. Unfairness score = inf.

---

**Subgroup 3**

If PREDICTOR RAT GANG AFFILIATION = 1, PREDICTOR RAT NARCOTIC ARRESTS = 2, PREDICTOR RAT TREND IN CRIMINAL ACTIVITY = (−8.200999999999999, −0.3], PREDICTOR RAT VICTIM SHOOTING INCIDENTS = 0:

Protected Subgroup = 'Black', 1.18% covered
  No recourses for this subgroup.
Protected Subgroup = 'White', 1.00% covered
  Make PREDICTOR RAT GANG AFFILIATION = 0, PREDICTOR RAT NARCOTIC ARRESTS = 0, PREDICTOR RAT TREND IN CRIMINAL ACTIVITY = (−8.200999999999999, −0.3], PREDICTOR RAT VICTIM SHOOTING INCIDENTS = 0 with effectiveness 31.82%
Bias against 'Black' due to Equal Cost of Effectiveness (threshold = 0.3). Unfairness score = inf.

Figure 6: Example of three Comparative Subgroup Counterfactuals in SSL; ref. Table 9

```
Subgroup 1
If age = 45−54, area = Unknown, parents = 1:
    Protected Subgroup = Male', 1.22% covered
        Make age=55−64, area=Rural with effectiveness 30.77%.
    Protected Subgroup = 'Female', 1.13% covered
        No recourses for this subgroup.
    Bias against Female due to Equal Cost of Effectiveness (threshold=0.3). Unfairness score = inf.
```

```
Subgroup 2
If age = 55−64, area = Unknown, homeowner = 1, income = Unknown, parents = 0, politics = Unknown, religion =
    Unknown:
    Protected Subgroup = 'Male', 2.53% covered
        Make homeowner=0, parents=1 with effectiveness 100.00%.
        Make homeowner=0, parents=1, religion=Christianity with effectiveness 100.00%.
        Make homeowner=0, parents=1, religion=Other with effectiveness 100.00%.
        Make area=Urban, parents=1, religion=Christianity with effectiveness 93.91%.
        Make area=Urban, parents=1, religion=Other with effectiveness 93.91%.
        Make homeowner=0, income=<100K, parents=1 with effectiveness 100.00%.
        Make area=Rural, parents=1, religion=Other with effectiveness 100.00%.
        Make area=Rural, parents=1, religion=Christianity with effectiveness 100.00%.
        Make homeowner=0, income=<100K, parents=1, religion=Christianity with effectiveness 100.00%.
        Make homeowner=0, income=<100K, parents=1, religion=Other with effectiveness 100.00%.
    Protected Subgroup = 'Female', 2.33% covered
        Make homeowner=0, parents=1 with effectiveness 100.00%.
        Make homeowner=0, parents=1, religion=Christianity with effectiveness 100.00%.
        Make homeowner=0, parents=1, religion=Other with effectiveness 100.00%.
        Make homeowner=0, income=<100K, parents=1 with effectiveness 100.00%.
        Make homeowner=0, income=<100K, parents=1, religion=Christianity with effectiveness 100.00%.
        Make homeowner=0, income=<100K, parents=1, religion=Other with effectiveness 100.00%.
    Bias against Female due to Equal Choice for Recourse (threshold=0.7). Unfairness score = 4.
```

```
Subgroup 3
If ages = 55−64, income = <100K, religion = Unknown:
    Protected Subgroup = 'Male', 1.02% covered
        Make religion=Christianity with effectiveness 0.00%.
        Make religion=Other with effectiveness 0.00%.
    Protected Subgroup = 'Female', 1.08% covered
        Make religion=Christianity with effectiveness 9.86%.
        Make religion=Other with effectiveness 9.86%.
    Bias against Male due to Equal Conditional Mean Recourse. Unfairness score = inf.
```

Figure 7: Example of three Comparative Subgroup Counterfactuals in Ad Campaign; ref. Table 10

# E  Further Discussion on Experiments

## E.1  Model Agnostic Results

We note that our method is model-agnostic and does not depend on the model class. Nonetheless, we have considered two additional models, XGBoost and a Neural Network, for the Adult dataset. We provide an example CSC for Subgroup 1 of Figure 3, one for the output of each model (XGBoost in Figure 8 and NN in Figure 9)

```
Subgroup 1
If  age=(41.0,  50.0], marital-status=Never-married, race=White, relationship=Not-in-family:
    Protected  Subgroup = 'Male',  1.13% covered
        No requerses for this subgroup.
    Protected  Subgroup = 'Female',  1.14% covered
        No requerses for this subgroup.
    No bias
```

Figure 8: Output of FACTS on an XGBoost model, for the same subgroup as Subgroup 1 of Figure 3.

```
Subgroup 1
If  age=(41.0,  50.0],  marital-status=Never-married,  race=White,  relationship=Not-in-family:
    Protected  Subgroup = Male',  1.21% covered
        No requerses for this subgroup.
    Protected  Subgroup = 'Female',  1.44% covered
        Make marital-status=Married-civ-spouse, relationship=Married with effectiveness 72.41%.
    Bias against 'Male' due to Equal Cost of Effectiveness (threshold = 0.7). Unfairness score = inf.
```

Figure 9: Output of FACTS on a simple NN model, for the same subgroup as Subgroup 1 of Figure 3.

## E.2  Generalization to multiclass demographic groups

The generalization of the algorithm to multiple protected groups (comprising multiclass demographic groups) is a straightforward process. For instance, we can compare the burden of each protected group against all other groups using a one-vs-all approach. An example of this methodology applied to the Adult dataset and its multi-valued race attribute is illustrated in Figure 10.

```
If  capital-gain=0, marital-status=Divorced:
    Protected  Subgroup = 'Asian-Pac-Islander',  12.95% covered
        Make capital-gain=15024, marital-status=Married-civ-spouse with effectiveness 100.00%.
        Make capital-gain=7688, marital-status=Married-civ-spouse with effectiveness 75.00%.
        Make capital-gain=7298, marital-status=Married-civ-spouse with effectiveness 75.00%.
    Protected  Subgroup = 'Amer-Indian-Eskimo',  20.37% covered
        Make capital-gain=15024, marital-status=Married-civ-spouse with effectiveness 100.00%.
        Make capital-gain=7688, marital-status=Married-civ-spouse with effectiveness 81.82%.
        Make capital-gain=7298, marital-status=Married-civ-spouse with effectiveness 72.73%.
    Protected  Subgroup = 'Black',  15.15% covered
        Make capital-gain=15024, marital-status=Married-civ-spouse with effectiveness 98.90%.
    Protected  Subgroup = 'Other',  12.97% covered
        Make capital-gain=15024, marital-status=Married-civ-spouse with effectiveness 100.00%.
        Make capital-gain=7688, marital-status=Married-civ-spouse with effectiveness 76.92%.
        Make capital-gain=7298, marital-status=Married-civ-spouse with effectiveness 69.23%.
    Protected  Subgroup = 'White',  16.96% covered
        Make capital-gain=15024, marital-status=Married-civ-spouse with effectiveness 99.41%.
        Make capital-gain=7688, marital-status=Married-civ-spouse with effectiveness 73.90%.
        Make capital-gain=7298, marital-status=Married-civ-spouse with effectiveness 70.43%.
    Bias against 'Black' due to Equal Choice of Recourse. Unfairness score = 2.
```

Figure 10: Output of FACTS for Adult for a multi-valued protected attribute (race); all protected subgroups other than 'Black' have three options (actions) to achieve recourse with an effectiveness of at least 0.7, while 'Black' has a single option. Therefore we conclude bias against the 'Black' using the Equal Choice of Recourse.

## E.3  Runtime and statistics

In Table 11, we present the runtime of FACTS for all datasets. We also provide statistics on the output generated by fp-growth for all these datasets, offering preliminary insights into scalability.

Additionally, we include the runtime for the Adult dataset, with sex serving as the protected attribute, for various combinations of minimum support threshold and the number of bins in the numerical attributes (see Table 13). These results encompass both the total runtime of FACTS and the time taken by FACTS in the assessment of fairness step. In Table 14 we report the corresponding statistics from fp-growth for the same combinations as presented in 13. It is worth noting that our primary contribution is conceptual in nature, focusing on the formalization of recourse unfairness notions, rather than technical, i.e., how to efficiently determine a good set of actions to explore.

Table 11: Runtime for all datasets (min-support threshold: 0.01 for Adult, COMPAS, SSL, Ad campaign; 0.04 for German credit)

|  | Adult (sex) | Adult (race) | COMPAS | SSL | Ad campaign | German credit |
|---|---|---|---|---|---|---|
| **Runtime (seconds)** | 1,875 | 2,208 | 11 | 1,807 | 860 | 689 |

Table 12: Statistics on the output of fp-growth for all datasets

|  | Adult (sex) | Adult (race) | COMPAS | SSL | Ad campaign | German credit |
|---|---|---|---|---|---|---|
| # male (white) subgroups (affected) | 27,510 | 27,767 | 1431 | 7,836 | 1,658 | 244,777 |
| # female (non-white) subgroups (affected) | 28,176 | 24,664 | 3,984 | 7,801 | 1,700 | 241,287 |
| # actions (unaffected) | 57,008 | 56,394 | 2,895 | 17,270 | 3,476 | 83,631 |
| # common subgroups | 13,300 | 18,692 | 95 | 6,552 | 1,651 | 36,970 |
| # valid actions | 97,413 | 112,879 | 188 | 86,782 | 4,274 | 117,143 |
| # subgroups | 12,556 | 15,672 | 88 | 6,551 | 1,280 | 31,509 |

Table 13: Runtime for Adult (sex) while varying min-support threshold and number of bins

| (min-support threshold, # bins) | (0.1, 10) | (0.05, 10) | (0.01, 10) | (0.05, 5) | (0.05, 20) |
|---|---|---|---|---|---|
| **Runtime (seconds)** | 35 | 178 | 3,539 | 265 | 155 |
| **Runtime to assess fairness (seconds)** | 8 | 10 | 17 | 10 | 10 |

Table 14: Statistics on the output of fp-growth for Adult (sex) while varying min-support threshold and number of bins

| (min-support threshold, # bins) | (0.1,10) | (0.05,10) | (0.01,10) | (0.05,5) | (0.05,20) |
|---|---|---|---|---|---|
| **# of male subgroups (affected)** | 997 | 3,017 | 25,569 | 4,017 | 2,309 |
| **# of female subgroups (affected)** | 933 | 3,024 | 26,173 | 4,021 | 2,397 |
| **# of common subgroups** | 453 | 1,583 | 12,093 | 1,969 | 1,126 |
| **# of actions (unaffected)** | 2,828 | 7,777 | 60,006 | 10,709 | 5,961 |
| **# of all valid actions** | 351 | 3,120 | 87,687 | 4,348 | 2,538 |
| **# of subgroups** | 283 | 1,235 | 10,975 | 1,706 | 887 |

# F Comparison of Fairness Metrics

The goal of this section is to answer the question: "How different are the fairness of recourse metrics". To answer this, we consider all subgroups and compare how they rank in terms of unfairness according to 12 distinct metrics. The results justify our claim in the main paper that the fairness metrics capture different aspects of recourse unfairness. For each dataset and protected attribute, we provide (a) the ranking analysis table, and (b) the aggregated rankings table.

The first column of the *ranking analysis* table shows the number of the most unfair subgroups per metric, i.e., how many ties are in rank 1. Depending on the unit of the unfairness score being compared between the protected subgroups (namely: cost, effectiveness, or number of actions), the number of ties can vary greatly. Therefore, we expect to have virtually no ties when comparing effectiveness percentages and to have many ties when comparing costs. The second and third columns show the number of subgroups where we observe bias in one direction (e.g., against males) and the opposite (e.g., against females) among the top 10% most unfair subgroups.

The *aggregated rankings* table is used as evidence that different fairness metrics capture different types of recourse unfairness. Each row concerns the subgroups that are the most unfair (i.e., tied at rank 1) according to each fairness metric. The values in the row indicate the average percentile ranks of these subgroups (i.e., what percentage of subgroups are more unfair) when ranked according to the other fairness metrics, shown as columns. Concretely, the value $v$ of each cell $i, j$ of this table is computed as follows:

1. We collect all subgroups of the fairness metric appearing in row $i$ that are ranked first (the most biased) due to this metric.

2. We compute the average ranking $a$ of these subgroups in the fairness metric appearing in column $j$.

3. We divide $a$ with the largest ranking tier of the fairness metric of column $j$ to arrive at $v$.

Each non-diagonal value of this table represents the relative ranking based on the specific metric of the column for all the subgroups that are ranked first in the metric of the respective row (all diagonal values of this table are left empty). A relative ranking of $v$ in a specific metric $m$ means that the most unfair subgroups of another metric are ranked lower on average (thus are fairer) for metric $m$.

## F.1 Comparison on Adult for protected attribute gender

The number of affected individuals in the test set for the adult dataset is 10,205. We first split the affected individuals on the set of affected males $D_1$ and the set of affected females $D_0$. The number of subgroups formed by running fp-growth with support threshold $1\%$ on $D_1$ and on $D_0$ and computing their intersection is 12,880. Our fairness metrics will evaluate and rank these subgroups based on the actions applied.

Tables 15 and 16 present the ranking analysis and the aggregated rankings respectively on the gender attribute, on the Adult dataset. Next, we briefly discuss the findings from these two tables; similar findings stand for the respective tables of the other datasets, thus we omit the respective discussion.

It is evident from Table 15 that the different ways to produce ranking scores by different definitions can lead to considerable differences in ties, i.e., the number of subgroups receiving the same rank (here only rank 1 is depicted). The "Top 10%" columns demonstrate interesting statistics on the protected subgroup for which bias is identified: while it is expected that mostly bias against "Female" will be identified, subgroups with reverse bias (bias against "Male") are identified, indicating robustness to gerrymandering, as hinted in Section 4 of the main paper.

Table 16 is produced to provide stronger evidence on the unique utility of the various presented definitions (see footnote 1 of the main paper: *"Additional examples, as well as statistics measuring this pattern on a significantly larger sample, are included in the supplementary material to further support this finding."*). In particular, in this table, for all subgroups that are ranked first in a definition, we calculate their average relative (normalized in $[0, 1]$) ranking in the remaining definitions. Given this, a value close to 1 means very low average rank and a value close to 0 means very high rank. Consequently, values away from 0 indicate the uniqueness and non-triviality of the different definitions and this becomes evident from the majority of the values of the table.

Table 15: Ranking Analysis in Adult (protected attribute gender)

| | # Most Unfair Subgroups | # Subgroups w. Bias against Males (in Top 10% Unfair Subgroups) | # Subgroups w. Bias against Females (in Top 10% Unfair Subgroups) |
|---|---|---|---|
| (Equal Cost of Effectiveness(Macro), 0.3) | 1673 | 56 | 206 |
| (Equal Cost of Effectiveness(Macro), 0.7) | 301 | 26 | 37 |
| (Equal Choice for Recourse, 0.3) | 2 | 54 | 286 |
| (Equal Choice for Recourse, 0.7) | 6 | 31 | 50 |
| Equal Effectiveness | 1 | 39 | 1040 |
| (Equal Effectiveness within Budget, 5.0 | 1 | 41 | 616 |
| (Equal Effectiveness within Budget, 10.0) | 1 | 6 | 904 |
| (Equal Effectiveness within Budget, 18.0) | 1 | 22 | 964 |
| (Equal Cost of Effectiveness(Micro), 0.3) | 1523 | 10 | 226 |
| (Equal Cost of Effectiveness(Micro), 0.7) | 290 | 38 | 27 |
| Equal(Conditional Mean Recourse) | 764 | 540 | 565 |
| (Fair Effectiveness-Cost Trade-Off, value) | 1 | 61 | 1156 |

Table 16: Aggregated Rankings in Adult (protected attribute gender)

| | (Equal Cost of Effectiveness (Macro), 0.3) | (Equal Cost of Effectiveness (Macro), 0.7) | (Equal Choice for Recourse, 0.3) | (Equal Choice for Recourse, 0.7) | Equal Effectiveness | (Equal Effectiveness within Budget, 5.0) | (Equal Effectiveness within Budget, 10.0) | (Equal Effectiveness within Budget, 18.0) | (Equal Cost of Effectiveness (Micro), 0.3) | (Equal Cost of Effectiveness (Micro), 0.7) | Equal(Conditional Mean Recourse) | (Fair Effectiveness-Cost Trade-Off, value) |
|---|---|---|---|---|---|---|---|---|---|---|---|---|
| (Equal Cost of Effectiveness(Macro), 0.3) | - | 1.0 | 0.836 | 1.0 | 0.214 | 0.509 | 0.342 | 0.285 | 0.3 | 1.0 | 0.441 | 0.237 |
| (Equal Cost of Effectiveness(Macro), 0.7) | 0.634 | - | 0.864 | 0.686 | 0.358 | 0.602 | 0.464 | 0.407 | 0.738 | 0.293 | 0.481 | 0.307 |
| (Equal Choice for Recourse, 0.3) | 0.018 | 1.0 | - | 1.0 | 0.001 | 0.006 | 0.001 | 0.001 | 0.017 | 1.0 | 0.105 | 0.001 |
| (Equal Choice for Recourse, 0.7) | 1.0 | 0.364 | 0.857 | - | 0.814 | 0.528 | 0.813 | 0.81 | 1.0 | 0.882 | 0.451 | 0.34 |
| Equal Effectiveness | 0.018 | 1.0 | 0.214 | 1.0 | - | 0.003 | 0.0 | 0.0 | 0.017 | 1.0 | 0.058 | 0.0 |
| (Equal Effectiveness within Budget, 5.0 | 0.018 | 1.0 | 0.857 | 1.0 | 0.006 | - | 0.004 | 0.006 | 0.017 | 1.0 | 1.0 | 0.006 |
| (Equal Effectiveness within Budget, 10.0) | 0.018 | 1.0 | 0.214 | 1.0 | 0.0 | 0.002 | - | 0.0 | 0.017 | 1.0 | 0.047 | 0.0 |
| (Equal Effectiveness within Budget, 18.0) | 0.018 | 1.0 | 0.214 | 1.0 | 0.0 | 0.003 | 0.0 | - | 0.017 | 1.0 | 0.058 | 0.0 |
| (Equal Cost of Effectiveness(Micro), 0.3) | 0.238 | 1.0 | 0.857 | 1.0 | 0.136 | 0.452 | 0.263 | 0.215 | - | 1.0 | 0.462 | 0.155 |
| (Equal Cost of Effectiveness(Micro), 0.7) | 0.611 | 0.279 | 0.864 | 0.771 | 0.336 | 0.621 | 0.449 | 0.402 | 0.7 | - | 0.465 | 0.295 |
| Equal(Conditional Mean Recourse) | 0.996 | 1.0 | 1.0 | 1.0 | 0.723 | 0.946 | 0.875 | 0.777 | 0.997 | 1.0 | - | 0.83 |
| (Fair Effectiveness-Cost Trade-Off, value) | 0.018 | 1.0 | 0.214 | 1.0 | 0.0 | 0.002 | 0.0 | 0.0 | 0.017 | 1.0 | 0.047 | - |

## F.2 Comparison on Adult for protected attribute race

The number of affected individuals in the test set for the adult dataset is 10,205. We first split the affected individuals on the set of affected whites $D_1$ and the set of affected non-whites $D_0$. The number of subgroups formed by running fp-growth with support threshold 1% on $D_1$ and on $D_0$ and computing their intersection is 16,621. Our fairness metrics will evaluate and rank these subgroups based on the actions applied.

Table 17: Ranking Analysis in Adult (protected attribute race)

| | # Most Unfair Subgroups | # Subgroups w. Bias against Whites (in Top 10% Unfair Subgroups) | # Subgroups w. Bias against Non-Whites (in Top 10% Unfair Subgroups) |
|---|---|---|---|
| (Equal Cost of Effectiveness(Macro), 0.3) | 1731 | 0 | 295 |
| (Equal Cost of Effectiveness(Macro), 0.7) | 325 | 7 | 51 |
| (Equal Choice for Recourse, 0.3) | 1 | 2 | 391 |
| (Equal Choice for Recourse, 0.7) | 2 | 10 | 60 |
| Equal Effectiveness | 1 | 6 | 1433 |
| (Equal Effectiveness within Budget, 1.15 | 1 | 50 | 24 |
| (Equal Effectiveness within Budget, 10.0) | 1 | 3 | 1251 |
| (Equal Effectiveness within Budget, 21.0) | 1 | 0 | 1423 |
| (Equal Cost of Effectiveness(Micro), 0.3) | 1720 | 0 | 294 |
| (Equal Cost of Effectiveness(Micro), 0.7) | 325 | 7 | 51 |
| Equal(Conditional Mean Recourse) | 2545 | 53 | 1316 |
| (Fair Effectiveness-Cost Trade-Off, value) | 2 | 0 | 0 |

## F.3 Comparison on COMPAS

The number of affected individuals in the test set for the COMPAS dataset is 745. We first split the affected individuals on the set of affected Caucasians $D_1$ and the set of affected African Americans $D_0$. The number of subgroups formed by running fp-growth with support threshold 1% on $D_1$ and on $D_0$ and computing their intersection is 995. Our fairness metrics will evaluate and rank these subgroups based on the actions applied.

## F.4 Comparison on SSL

The number of affected individuals in the test set for the SSL dataset is 11,343. We first split the affected individuals on the set of affected blacks $D_1$ and the set of affected whites $D_0$ based on the race attribute (appears with the name RACE CODE CD in the dataset). The number of subgroups

Table 18: Aggregated Rankings in Adult (protected attribute race)

| | (Equal Cost of Effectiveness(Macro), 0.3) | (Equal Cost of Effectiveness(Macro), 0.7) | (Equal Choice for Recourse, 0.3) | (Equal Choice for Recourse, 0.7) | Equal Effectiveness | (Equal Effectiveness within Budget, 1.15) | (Equal Effectiveness within Budget, 10.0) | (Equal Effectiveness within Budget, 21.0) | (Equal Cost of Effectiveness(Micro), 0.3) | (Equal Cost of Effectiveness(Micro), 0.7) | Equal(Conditional Mean Recourse) | (Fair Effectiveness-Cost Trade-Off, value) |
|---|---|---|---|---|---|---|---|---|---|---|---|---|
| (Equal Cost of Effectiveness(Macro), 0.3) | - | 1.0 | 0.845 | 1.0 | 0.162 | 0.996 | 0.283 | 0.177 | 0.026 | 1.0 | 0.448 | 0.194 |
| (Equal Cost of Effectiveness(Macro), 0.7) | 0.7 | - | 0.9 | 0.829 | 0.147 | 0.973 | 0.315 | 0.169 | 0.698 | 0.05 | 0.421 | 0.12 |
| (Equal Choice for Recourse, 0.3) | 0.419 | 1.0 | - | 1.0 | 1.0 | 1.0 | 0.03 | 1.0 | 0.419 | 1.0 | 0.782 | 0.073 |
| (Equal Choice for Recourse, 0.7) | 1.0 | 0.095 | 0.909 | - | 0.644 | 1.0 | 0.003 | 0.328 | 1.0 | 0.1 | 0.041 | 0.011 |
| Equal Effectiveness | 0.023 | 1.0 | 0.909 | 1.0 | - | 1.0 | 0.01 | 0.0 | 0.023 | 1.0 | 0.0 | 0.0 |
| (Equal Effectiveness within Budget, 1.15) | 1.0 | 0.048 | 1.0 | 0.857 | 0.069 | - | 0.047 | 0.07 | 1.0 | 0.05 | 1.0 | 0.102 |
| (Equal Effectiveness within Budget, 10.0) | 0.395 | 0.048 | 0.818 | 0.571 | 0.001 | 1.0 | - | 0.001 | 0.395 | 0.05 | 0.611 | 0.002 |
| (Equal Effectiveness within Budget, 21.0) | 0.023 | 1.0 | 0.909 | 1.0 | 0.0 | 1.0 | 0.01 | - | 0.023 | 1.0 | 0.0 | 0.0 |
| (Equal Cost of Effectiveness(Micro), 0.3) | 0.023 | 1.0 | 0.845 | 1.0 | 0.162 | 0.996 | 0.284 | 0.177 | - | 1.0 | 0.449 | 0.195 |
| (Equal Cost of Effectiveness(Micro), 0.7) | 0.7 | 0.048 | 0.9 | 0.829 | 0.147 | 0.973 | 0.315 | 0.169 | 0.698 | - | 0.421 | 0.12 |
| Equal(Conditional) Mean Recourse | 0.979 | 1.0 | 1.0 | 1.0 | 0.628 | 1.0 | 0.778 | 0.633 | 0.979 | 1.0 | - | 0.721 |
| (Fair Effectiveness-Cost Trade-Off, value) | 0.023 | 1.0 | 0.818 | 1.0 | 0.001 | 1.0 | 0.012 | 0.001 | 0.023 | 1.0 | 0.003 | - |

Table 19: Ranking Analysis in COMPAS

| | # Most Unfair Subgroups | # Subgroups w. Bias against Caucasians (in Top 10% Unfair Subgroups) | # Subgroups w. Bias against African-Americans (in Top 10% Unfair Subgroups) |
|---|---|---|---|
| (Equal Cost of Effectiveness(Macro), 0.3) | 51 | 0 | 11 |
| (Equal Cost of Effectiveness(Macro), 0.7) | 46 | 0 | 6 |
| (Equal Choice for Recourse, 0.3) | 13 | 12 | 8 |
| (Equal Choice for Recourse, 0.7) | 15 | 8 | 6 |
| Equal Effectiveness | 1 | 14 | 37 |
| (Equal Effectiveness within Budget, 1.0) | 4 | 16 | 30 |
| (Equal Effectiveness within Budget, 10.0) | 1 | 20 | 39 |
| (Equal Cost of Effectiveness(Micro), 0.3) | 51 | 0 | 11 |
| (Equal Cost of Effectiveness(Micro), 0.7) | 46 | 0 | 6 |
| Equal(Conditional Mean Recourse) | 37 | 19 | 24 |
| (Fair Effectiveness-Cost Trade-Off, value) | 5 | 18 | 62 |

Table 20: Aggregated Rankings in COMPAS

| | (Equal Cost of Effectiveness(Macro), 0.3) | (Equal Cost of Effectiveness(Macro), 0.7) | (Equal Choice for Recourse, 0.3) | (Equal Choice for Recourse, 0.7) | Equal Effectiveness | (Equal Effectiveness within Budget, 1.0) | (Equal Effectiveness within Budget, 10.0) | (Equal Cost of Effectiveness(Micro), 0.3) | (Equal Cost of Effectiveness(Micro), 0.7) | Equal(Conditional Mean Recourse) | (Fair Effectiveness-Cost Trade-Off, value) |
|---|---|---|---|---|---|---|---|---|---|---|---|
| (Equal Cost of Effectiveness(Macro), 0.3) | - | 1.0 | 0.65 | 1.0 | 0.169 | 0.801 | 0.398 | 0.2 | 1.0 | 0.797 | 0.226 |
| (Equal Cost of Effectiveness(Macro), 0.7) | 0.96 | - | 0.925 | 0.625 | 0.127 | 0.518 | 0.236 | 0.96 | 0.2 | 0.52 | 0.149 |
| (Equal Choice for Recourse, 0.3) | 0.32 | 0.76 | - | 0.775 | 0.082 | 1.0 | 0.178 | 0.32 | 0.76 | 0.297 | 0.116 |
| (Equal Choice for Recourse, 0.7) | 0.9 | 0.46 | 0.8 | - | 0.424 | 0.484 | 0.057 | 0.9 | 0.46 | 0.259 | 0.045 |
| Equal Effectiveness | 0.2 | 1.0 | 0.75 | 1.0 | - | 1.0 | 0.003 | 0.2 | 1.0 | 0.003 | 0.002 |
| (Equal Effectiveness within Budget, 1.0) | 0.8 | 1.0 | 0.75 | 0.75 | 0.0 | - | 1.0 | 0.8 | 1.0 | 0.413 | 0.002 |
| (Equal Effectiveness within Budget, 10.0) | 0.2 | 1.0 | 0.75 | 1.0 | 0.003 | 1.0 | - | 0.2 | 1.0 | 0.003 | 0.002 |
| (Equal Cost of Effectiveness(Micro), 0.3) | 0.2 | 1.0 | 0.65 | 1.0 | 0.169 | 0.801 | 0.398 | - | 1.0 | 0.797 | 0.226 |
| (Equal Cost of Effectiveness(Micro), 0.7) | 0.96 | 0.2 | 0.925 | 0.625 | 0.127 | 0.518 | 0.236 | 0.96 | - | 0.52 | 0.149 |
| Equal(Conditional) Mean Recourse | 0.98 | 1.0 | 1.0 | 1.0 | 0.507 | 0.772 | 0.512 | 0.98 | 1.0 | - | 0.627 |
| (Fair Effectiveness-Cost Trade-Off, value) | 0.68 | 1.0 | 0.75 | 0.8 | 0.801 | 0.202 | 0.801 | 0.68 | 1.0 | 0.331 | - |

formed by running fp-growth with support threshold 1% on $D_1$ and on $D_0$ and computing their intersection is 6,551. Our fairness metrics will evaluate and rank these subgroups based on the actions applied.

Table 21: Ranking Analysis in SSL

| | # Most Unfair Subgroups | # Subgroups w. Bias against Whites (in Top 10% Unfair Subgroups) | # Subgroups w. Bias against Blacks (in Top 10% Unfair Subgroups) |
|---|---|---|---|
| (Equal Cost of Effectiveness(Macro), 0.3) | 371 | 10 | 107 |
| (Equal Cost of Effectiveness(Macro), 0.7) | 627 | 26 | 124 |
| (Equal Choice for Recourse, 0.3) | 1 | 108 | 184 |
| (Equal Choice for Recourse, 0.7) | 16 | 78 | 229 |
| Equal Effectiveness | 1 | 15 | 389 |
| (Equal Effectiveness within Budget, 1.0) | 18 | 18 | 436 |
| (Equal Effectiveness within Budget, 2.0) | 2 | 19 | 532 |
| (Equal Effectiveness within Budget, 10.0) | 1 | 15 | 548 |
| (Equal Cost of Effectiveness(Micro), 0.3) | 458 | 5 | 135 |
| (Equal Cost of Effectiveness(Micro), 0.7) | 671 | 23 | 130 |
| Equal(Conditional Mean Recourse) | 100 | 41 | 434 |
| (Fair Effectiveness-Cost Trade-Off, value) | 80 | 76 | 544 |

Table 22: Aggregated Rankings in SSL

| | (Equal Cost of Effectiveness(Macro), 0.3) | (Equal Cost of Effectiveness(Macro), 0.7) | (Equal Choice for Recourse, 0.3) | (Equal Choice for Recourse, 0.7) | Equal Effectiveness | (Equal Effectiveness within Budget, 1.0) | (Equal Effectiveness within Budget, 2.0) | (Equal Effectiveness within Budget, 10.0) | (Equal Cost of Effectiveness(Micro), 0.3) | (Equal Cost of Effectiveness(Micro), 0.7) | Equal(Conditional Mean Recourse) | (Fair Effectiveness-Cost Trade-Off, value) |
|---|---|---|---|---|---|---|---|---|---|---|---|---|
| (Equal Cost of Effectiveness(Macro), 0.3) | - | 0.883 | 0.854 | 0.988 | 0.216 | 0.285 | 0.238 | 0.238 | 0.3 | 0.843 | 0.678 | 0.338 |
| (Equal Cost of Effectiveness(Macro), 0.7) | 0.929 | - | 0.877 | 0.725 | 0.239 | 0.421 | 0.332 | 0.264 | 0.871 | 0.314 | 0.829 | 0.342 |
| (Equal Choice for Recourse, 0.3) | 0.143 | 1.0 | - | 1.0 | 0.328 | 0.704 | 0.464 | 0.368 | 0.143 | 1.0 | 0.727 | 0.601 |
| (Equal Choice for Recourse, 0.7) | 1.0 | 0.167 | 0.769 | - | 0.083 | 0.177 | 0.127 | 0.086 | 1.0 | 0.143 | 0.926 | 0.135 |
| Equal Effectiveness | 0.143 | 0.167 | 0.923 | 0.938 | - | 0.002 | 0.0 | 0.0 | 0.143 | 0.143 | 0.0 | 0.003 |
| (Equal Effectiveness within Budget, 1.0) | 0.857 | 0.833 | 0.854 | 0.923 | 0.89 | - | 0.923 | 0.876 | 0.857 | 0.857 | 0.327 | 0.0 |
| (Equal Effectiveness within Budget, 2.0) | 0.286 | 0.333 | 0.923 | 0.938 | 0.5 | 0.002 | - | 0.5 | 0.286 | 0.286 | 0.0 | 0.003 |
| (Equal Effectiveness within Budget, 10.0) | 0.143 | 0.167 | 0.923 | 0.938 | 0.0 | 0.002 | 0.0 | - | 0.143 | 0.143 | 0.0 | 0.003 |
| (Equal Cost of Effectiveness(Micro), 0.3) | 0.443 | 0.833 | 0.877 | 0.969 | 0.143 | 0.312 | 0.198 | 0.154 | - | 0.843 | 0.729 | 0.268 |
| (Equal Cost of Effectiveness(Micro), 0.7) | 0.9 | 0.383 | 0.892 | 0.788 | 0.203 | 0.406 | 0.299 | 0.225 | 0.886 | - | 0.816 | 0.327 |
| Equal(Conditional) Mean Recourse | 0.6 | 0.733 | 0.946 | 0.969 | 0.244 | 0.464 | 0.395 | 0.378 | 0.514 | 0.729 | - | 0.396 |
| (Fair Effectiveness-Cost Trade-Off, value) | 0.971 | 0.967 | 0.838 | 0.869 | 0.967 | 0.774 | 0.977 | 0.96 | 0.971 | 0.971 | 0.837 | - |

## F.5 Comparison on Ad Campaign

The number of affected individuals in the test set for the Ad campaign dataset is 273,773. We first split the affected individuals on the set of affected males $D_1$ and the set of affected females $D_0$ based on the gender attribute. The number of subgroups formed by running fp-growth with support threshold $1\%$ on $D_1$ and on $D_0$ and computing their intersection is 1,432. Our fairness metrics will evaluate and rank these subgroups based on the actions applied.

Table 23: Ranking Analysis in Ad Campaign

| | # Most Unfair Subgroups | # Subgroups w. Bias against Males (in Top 10% Unfair Subgroups) | # Subgroups w. Bias against Females (in Top 10% Unfair Subgroups) |
|---|---|---|---|
| (Equal Cost of Effectiveness(Macro), 0.3) | 427 | 0 | 44 |
| (Equal Cost of Effectiveness(Macro), 0.7) | 264 | 0 | 26 |
| (Equal Choice for Recourse, 0.3) | 2 | 1 0 | 66 |
| (Equal Choice for Recourse, 0.7) | 384 | 0 | 39 |
| Equal Effectiveness | 15 | 0 | 123 |
| (Equal Effectiveness within Budget, 1.0) | 1 | 0 | 42 |
| (Equal Effectiveness within Budget, 5.0) | 10 | 0 | 114 |
| (Equal Cost of Effectiveness(Micro), 0.3) | 427 | 0 | 44 |
| (Equal Cost of Effectiveness(Micro), 0.7) | 264 | 0 | 26 |
| Equal(Conditional Mean Recourse) | 108 | 9 | 74 |
| (Fair Effectiveness-Cost Trade-Off, value) | 15 | 0 | 128 |

Table 24: Aggregated Rankings in Ad Campaign

| | (Equal Cost of Effectiveness (Macro), 0.3) | (Equal Cost of Effectiveness (Macro), 0.7) | (Equal Choice for Recourse, 0.3) | (Equal Choice for Recourse, 0.7) | Equal Effectiveness | (Equal Effectiveness within Budget, 1.0) | (Equal Effectiveness within Budget, 5.0) | (Equal Cost of Effectiveness (Micro), 0.3) | (Equal Cost of Effectiveness (Micro), 0.7) | Equal(Conditional Mean Recourse) | (Fair Effectiveness-Cost Trade-Off, value) |
|---|---|---|---|---|---|---|---|---|---|---|---|
| (Equal Cost of Effectiveness(Macro), 0.3) | - | 0.7 | 0.483 | 0.6 | 0.167 | 1.0 | 0.276 | 0.25 | 0.7 | 0.487 | 0.154 |
| (Equal Cost of Effectiveness(Macro), 0.7) | 0.25 | - | 0.35 | 0.333 | 0.082 | 1.0 | 0.21 | 0.25 | 0.5 | 0.506 | 0.079 |
| (Equal Choice for Recourse, 0.3) | 0.25 | 1.0 | - | 1.0 | 0.73 | 1.0 | 1.0 | 0.25 | 1.0 | 0.037 | 0.338 |
| (Equal Choice for Recourse, 0.7) | 0.5 | 0.65 | 0.333 | - | 0.296 | 0.851 | 0.385 | 0.5 | 0.65 | 0.566 | 0.273 |
| Equal Effectiveness | 0.25 | 0.5 | 0.333 | 0.333 | - | 1.0 | 0.205 | 0.25 | 0.5 | 0.002 | 0.001 |
| (Equal Effectiveness within Budget, 1.0) | 1.0 | 1.0 | 1.0 | 1.0 | 0.714 | - | 0.671 | 1.0 | 1.0 | 0.305 | 0.395 |
| (Equal Effectiveness within Budget, 5.0) | 0.25 | 0.7 | 0.333 | 0.333 | 0.001 | 1.0 | - | 0.25 | 0.7 | 0.002 | 0.001 |
| (Equal Cost of Effectiveness(Micro), 0.3) | 0.25 | 0.5 | 0.483 | 0.6 | 0.167 | 1.0 | 0.276 | - | 0.7 | 0.487 | 0.154 |
| (Equal Cost of Effectiveness(Micro), 0.7) | 0.25 | 0.5 | 0.35 | 0.333 | 0.082 | 1.0 | 0.21 | 0.25 | - | 0.506 | 0.079 |
| Equal(Conditional) Mean Recourse | 0.525 | 0.75 | 0.65 | 0.7 | 0.25 | 1.0 | 0.408 | 0.525 | 0.75 | - | 0.267 |
| (Fair Effectiveness-Cost Trade-Off, value) | 0.25 | 0.5 | 0.333 | 0.333 | 0.001 | 1.0 | 0.205 | 0.25 | 0.5 | 0.002 | - |

# G  Discussion of Limitations & Responsible Usage Guidelines

Determining costs for specific counterfactuals poses numerous challenges and constraints. The associated cost functions are intricate, dataset-specific, and frequently require the expertise of a domain specialist. These cost functions can depend on individual characteristics or human interpretation of the perceived difficulty of specific changes, potentially giving rise to concerns like breaches of user privacy (as the expert may require access to all individual user characteristics [25]) or potential manipulation of cost functions by malicious specialists to conceal existing biases.

Recognizing the difficulties of finding the dataset-dependent cost functions, we implement various "natural" cost functions tailored to different feature types (e.g., $L_1$ norm, exponential distances, etc.). These serve only as suggestions to the expert using our framework, are susceptible to change, and can be customized for specific datasets. While the creation of precise cost models is an interesting research area, it is not our primary focus. Instead, our primary goal is to identify a specific type of bias (difficulty to achieve recourse) towards specific subpopulations. Therefore, we have introduced fairness metrics that are independent of cost considerations, recognizing the aforementioned inherent difficulties in defining costs. We recognize that a single metric, whether cost-dependent or not, cannot identify all forms of bias that may be present within a model's predictions. We do not aim to replace the existing statistical measures for bias detection, rather than complement them and capture possible gaps, as described in the introduction of our paper and Section 5.

Our framework evaluates subgroups based on a range of fairness metrics, identifying 'candidate' subgroups that may experience unfair treatment under scrutiny. While the number of such 'candidate' subgroups can be substantial across different metrics, this abundance can serve as a safeguard against malicious users, as cost-oblivious rules are immune to user influence. We emphasize our efforts to produce comprehensive summaries that are easily understandable for the end-users and help them make informative decisions regarding the existence of the groups that are unfairly treated.

The ultimate decision regarding which of the 'candidate' subgroups are considered unfairly treated rests with the expert. Our comprehensive summaries highlight the top-scoring 'candidate' subgroups across multiple metrics, with the ultimate goal of helping the end users and deterring malicious users from deliberately ignoring the framework's suggestions. Explanations accompany FACTS' suggestions, requiring users to justify their rejection and serving as a barrier against deliberate malicious actions. While our summaries are interpretable and customizable, we plan to enhance our framework with visualizations in future work to further assist end-users. A user study would also be helpful to assess FACTS' interpretability to the end-user.

It is important to note that FACTS, like other resource generation algorithms, may be susceptible to errors, as demonstrated in prior research [31]. The outcomes produced by FACTS can vary based on the chosen configuration, including hyperparameters and metric selection. These variations can either obscure biases present within a classifier or indicate their absence. It is crucial to emphasize that since FACTS is an explainable framework for bias assessment within subgroups, its primary purpose is to serve as a guidance tool for auditors to identify subgroups requiring in-depth analysis of fairness criteria violations. Responsible usage of our framework necessitates transparency regarding the chosen configuration. Therefore, we recommend running FACTS with different hyperparameters and utilizing as many metrics as possible.

Another potential limitation relates to GDPR and similar regulations, which may restrict access to protected attributes (confidential information, etc.) even in bias detection applications. It's important to clarify that our framework maintains privacy and avoids personal information leakage. Only summaries of results are presented to the end-users, limited to subgroups that impose the minimum size requirement. These features could be further enhanced by adding restrictions to the size requirements if legislation or standardized requirements existed. In essence, our approach addresses complex challenges in bias detection, while upholding fairness, respecting privacy requirements, and ensuring interpretability for end-users.

An acknowledged constraint of our framework pertains to how we generate actions, as explained in Section 3. Both actions and subgroups are derived using a frequent itemset mining algorithm. We have consciously opted against altering our approach to action generation, to minimize computational complexity and enhance action effectiveness. Importantly, the same actions are applied to all groups sharing a common predicate during the computation of fairness metrics.

