# OpenReview forum: "Fairness Aware Counterfactuals for Subgroups"
_NeurIPS.cc/2023/Conference — NeurIPS 2023 poster_

### Official Review · Reviewer_ZFdX · 2023-07-04

**Soundness:** 3 good
**Presentation:** 3 good
**Contribution:** 4 excellent
**Rating:** 7
**Confidence:** 4

**Summary:**

The authors identify various aspects of fairness that require consideration when assessing recourse bias between subgroups. They propose FACTS (Fairness Aware Counterfactuals for Subgroups), in an attempt to lay foundations for a framework that can be used to audit subgroup fairness through counterfactual explanations (CEs). Experimental evaluation is relatively well thought out and thorough.

**Strengths:**

1. The paper is easy to read, with a logical structure and natural flow. In most cases, relevant examples are provided where necessary.

2. I believe the issue being addressed is complex, and hard conclusions regarding the effectiveness of fairness tools should be made sparingly. However, the paper does an excellent job of analysing the problem at a deeper level than that which currently exists in the literature [1]. Explicit separation between the micro and macro viewpoint is well positioned.

3. While there are still gaps to be filled, this paper lays a strong foundation for future research into the use of counterfactuals in assessing recourse bias/unfairness. It is worth noting that [2] conducts similar analyses of the potential pitfalls of bias assessment via subgroup comparison, arriving at a similar conclusion to this paper that recourses should be compared 1 to 1 for more reliable results.

[1] Michael Kearns, Seth Neel, Aaron Roth, and Zhiwei Steven Wu. Preventing fairness gerrymandering: Auditing and learning for subgroup fairness. ICML 2018.

[2] Dan Ley, Saumitra Mishra, Daniele Magazzeni. GLOBE-CE: A Translation-Based Approach for Global Counterfactual Explanations. ICML 2023.


**Weaknesses:**

I appreciate the thoroughness of the authors in tackling an issue as complex as this. There are a few important gaps that I believe may have evaded their attention, which I will detail below.

1. The assumption that while cost is not exactly quantifiable, a given action's cost is uniform across the input space, and can thus be compared between instances, does not always hold. One simple example would be changing salary by amount X. Individuals commanding higher initial salaries are likely to find this action easier. I would thus modify the "oblivious to the cost function" claim appropriately (though it is still a more than reasonable starting point for now).

2. Evaluation of fairness appears quite sensitive to the set of actions A chosen (specifically the fixed cost used/described in lines 290-302). Such a budget can often handle poorly the asymmetries between subgroup cost distributions for a given action. That said, multiple cost budgets are evaluated in the experiments, to provide a more complete overview. It's also worth considering the idea of scaling the cost of certain action directions, as in [2], and evaluating fairness accordingly.

3. In a similar vein, frequent itemsets are prone to resolution issues with numerical features. Moving beyond the apriori approach in [3] is wise, though frequent itemsets often do not uncover the minimal cost directions to flip predictions. Additionally, if the motivation behind fp-growth is primarily efficiency, it would be useful to have some analysis of this.

4. Discussion of when and how the fairness metrics can be manipulated (as above) is needed. Additionally, in practical settings, individuals are each in control of diverse actions, often unique to them. The effect of such actions (not included in but confounding with the feature space) is worth pointing out.

5. A large quantity of metrics are introduced. Table 1 caption could do with more clarification so readers don't have to study the text to understand the highlighted values.

[3] Kaivalya Rawal and Himabindu Lakkaraju. Beyond individualized recourse: Interpretable and interactive summaries of actionable recourses. NeurIPS 2020.

**Questions:**

The bins chosen for numerical features are rather large, and so may not effectively capture the minimum cost recourses between subgroups if the numerical features are influential, which might commonly be the case. This is one of the issues with frequent itemset mining, and something that [3] originally suffered with- do the authors have any proposed solutions? From my experience in the area, this is a fairly serious shortcoming, and something that prevented me from awarding this paper an even higher score.

**Limitations:**

Described above. Limitations associated with choosing parameters for fairness metrics and determining appropriate action spaces should be discussed in the revision.

The research contributes valuable insights and provides a robust foundation for future work, though I would suggest to keep in mind the inherent complexity of the fairness problem, which is indeed emphasised in the paper. Overconfidence in the reliability of the proposed framework may not encourage an appropriately critical approach to this multifaceted issue i.e. conclusions drawn should be cautiously optimistic and mindful of the above limitations.

---

> ### Author Rebuttal · Authors · 2023-08-10
>
> **W1 "The assumption that while cost is not exactly quantifiable, a given action's cost is uniform across the input space, and can thus be compared between instances, does not always hold. One simple example would be changing salary by amount X. Individuals commanding higher initial salaries are likely to find this action easier."**
>
> We thank the reviewer for raising this point, which only concerns the cost-oblivious fairness metrics. There, we will make this assumption explicit: the cost of a given action $a$ (e.g., increase salary by amount $\delta$) is the same for all individuals $x \neq x'$, i.e., $cost(a, x) = cost(a, x')$.
> When this is not the case, we can think of two alternatives that are straightforward to implement in FACTS:
> - We can keep the subpopulation bins for that attribute small (or adjust them accordingly) so that the assumption approximately holds for each subpopulation.
> - Adapt the definition of actions so as to satisfy the assumption. E.g., for salary, we consider actions that increase salary by $\gamma$%.
>
>
> **W2 "Evaluation of fairness appears quite sensitive to the set of actions A chosen (specifically the fixed cost used/described in lines 290-302). Such a budget can often handle poorly the asymmetries between subgroup cost distributions for a given action. That said, multiple cost budgets are evaluated in the experiments, to provide a more complete overview. It's also worth considering the idea of scaling the cost of certain action directions, as in [A], and evaluating fairness accordingly"**
>
> We emphasize that our goal is to accurately capture the real effectiveness-cost distribution (ecd) within a subpopulation. We achieve this by considering the cost and effectiveness of all valid actions among a set $A$. Augmenting this set with additional actions can help improve the accuracy of the ecd. As suggested, for each action $a \in A$ we may also consider other actions derived from $a$ by scaling $a$'s numerical part; e.g., if $a$ says change income from [40K,50K] to [50K,60K], i.e., add 10K, we can also consider actions that change the income by 5K, 10K, 15K, for example. This can also be achieved by applying a different numerical binning for the actions as we discuss in the author rebuttal. Another approach would be to sample the space of possible actions uniformly at random.
>
>
> **W3 "In a similar vein, frequent itemsets are prone to resolution issues with numerical features. Moving beyond the apriori approach in [3] is wise, though frequent itemsets often do not uncover the minimal cost directions to flip predictions."**
>
> We want to emphasize that all model-agnostic methods that explore a finite set of actions $A$ have an inherent limitation to how well they can discover the minimum recourse cost. Nonetheless, as discussed in the previous comment, we can foresee several ways to augment the set $A$ so as to most accurately represent the ecd.
>
>
> **W3 "Additionally, if the motivation behind fp-growth is primarily efficiency, it would be useful to have some analysis of this."**
>
> Tables 1-4 in the global response show the runtime of our fp-growth-based generation of subpopulations and actions, and also provides some scalability results. We note however that our main contribution is a conceptual one, i.e., how to formalize notions of recourse unfairness, rather that a technical one, i.e., how to efficiently determine a good set of actions to explore.
>
>
> **W4 "Discussion of when and how the fairness metrics can be manipulated (as above) is needed. Additionally, in practical settings, individuals are each in control of diverse actions, often unique to them. The effect of such actions (not included in but confounding with the feature space) is worth pointing out."**
>
> We thank the reviewer for raising these issues; we intend to discuss them in greater detail in a limitation section. The manipulability of the method and the robustness of the conclusions drawn is an important issue. We note that while local/individual counterfactuals have been found to be prone to manipulability, our fairness auditing method is more robust as it considers multiple counterfactuals for the same individual and draws conclusions at a subpopulation level. Clearly, more research in this direction is called for.
>
> Regarding the last observation, this boils down to how well the cost function models the world; e.g., it could capture causal structures in the domain, and personalized/unique actions/costs.
>
>
> **Q1 "The bins chosen for numerical features are rather large, and so may not effectively capture the minimum cost recourses between subgroups if the numerical features are influential, which might commonly be the case. This is one of the issues with frequent itemset mining, and something that [30] originally suffered with- do the authors have any proposed solutions? From my experience in the area, this is a fairly serious shortcoming, and something that prevented me from awarding this paper an even higher score."**
>
> We note the distinction between binning for subpopulations and binning for actions, which can be done separately as discussed in the global response, and has the potential to define a more fine-grained action set $A$. Additional ways to augment or to define a better set $A$ without relying on frequent itemset mining are also outlined in the response to **W2**, but are somewhat orthogonal to our main contribution. Certainly, the implementation of our auditing framework can benefit from recent advances in the field like [A].

---

> > ### Comment · Reviewer_ZFdX · 2023-08-17
> > **Response to authors**
> >
> > I would like to thank the authors for their time spent in addressing my concerns. While the response is mostly satisfying, I maintain my score of 7 and vote for acceptance of this work.

---

### Official Review · Reviewer_EeTT · 2023-07-04

**Soundness:** 3 good
**Presentation:** 3 good
**Contribution:** 2 fair
**Rating:** 6
**Confidence:** 3

**Summary:**

The paper presents a framework called FACTS (Fairness Aware Counterfactuals for Subgroups) for auditing subgroup fairness through counterfactual explanations. The authors aim to formulate different aspects of the difficulty individuals face in achieving recourse, either at the micro level (individuals within subgroups) or at the macro level (subgroups as a whole). They introduce new notions of subgroup fairness that are robust and provide an efficient, model-agnostic, and explainable framework for evaluating subgroup fairness. The authors demonstrate the advantages and wide applicability of their approach through an experimental evaluation using benchmark datasets.

**Strengths:**

+ The paper addresses an important and timely topic in machine learning, namely fairness in decision-making processes.
+ The authors propose a novel framework, FACTS, for auditing subgroup fairness, which is model-agnostic and highly parameterizable.
+ The paper provides a thorough explanation of the different notions of subgroup fairness and their implications.
+ The experimental evaluation demonstrates the effectiveness and efficiency of the proposed approach on benchmark datasets.
+ The paper is well written and organized.

**Weaknesses:**

There are a few weaknesses:

- The paper could benefit from a more detailed description of the methodology used in the experimental evaluation. While the authors have provided an overview of the experimental setup and the data collection process, a more comprehensive explanation of the specific steps taken would greatly enhance the clarity and reproducibility of the study. For example, providing information about the specific criteria used to select the sources or the process of extracting and preprocessing the data would be useful for identifying the potential data generating bias.

- In addition, providing more information on the statistical analysis performed on the experimental results would enhance the credibility of the findings. The authors mainly use Comparative Subgroup Counterfactuals for evaluating. However, it remains unclear whether the observations are significant, and how do they generalize to high-dimensional data as well.

- The framework may face challenges in cases where the protected attributes are not well-defined or are subject to interpretation. How does FACTS perform when the underlying sensitive attributes are unknown (which is esp. true for many applications)?

**Questions:**

Please refer to the weakness section for main questions. In addition to points raised above, there are a few questions remained:

- How does the proposed framework handle the issue of defining a cost function for recourse?
- Are there any computational or scalability limitations of the FACTS framework?
- How does the performance of the FACTS framework compare to other state-of-the-art approaches for auditing subgroup fairness?
- The proposed framework relies on the assumption that counterfactual explanations can accurately capture the underlying causes of bias and provide actionable insights.
- The effectiveness of the FACTS framework may vary depending on the specific dataset and domain.
- The framework may face challenges in cases where the protected attributes are not well-defined or are subject to interpretation.

**Limitations:**

The main paper does not have a limitation statement, although the authors mentioned a few in the conclusion. There's no broader societal impact statement about the potential negative societal impact.

---

> ### Author Rebuttal · Authors · 2023-08-10
>
> **W1 "more detailed description of the methodology used"**
>
> We acknowledge that some details about the experimental methodology description are missing in the main text, and probably not adequately covered in the supplementary material. We intend to remedy this should the paper be accepted. We should note however, that standard preprocessing and train-test practices were applied on all datasets, which are mostly widely used datasets for benchmarking fairness methods. Finally, we note that we have included more detailed information for the additional experiments we have conducted during this rebuttal phase, and which we present in the pdf attached to the author rebuttal.
>
>
> **W2 "more information on the statistical analysis performed on the experimental results" "whether the observations are significant, and how do they generalize to high-dimensional data"**
>
> We note that the experimental results are derived by applying the steps outlined in Section 3 and then applying the definitions of Section 2.3 in a straightforward manner. There is no statistical post-processing to include or exclude data points.
>
> High-dimensional data may lead to subgroups with few individuals. Please refer to our response to comment *Q3* of *Reviewer fCWq* on significance.
>
>
> **W3 "How does FACTS perform when the underlying sensitive attributes are unknown?"**
>
> Similar to fairness literature (see Fliptest, Fairtest, AReS, Globe-CE), the problem formulation that FACTS adopts is that the protected attributes are known in advance. Techniques that discover protected attributes is a very interesting but orthogonal research direction; i.e., such techniques can be applied prior to running FACTS.
>
> Nonetheless, we wish to draw attention to two important aspects of FACTS.
>
> First, FACTS can investigate whether recourse bias manifests in intersectional subgroups, e.g. black men of certain age groups, despite being absent when comparing all black with all white men, for example.
>
> Second, FACTS is not restricted to apply solely on protected attributes. Any attribute (and even a multiclass/multivalued attribute) can be treated as "protected" and given as input to FACTS. FACTS then would assess any disparities in recourse across subpopulations defined by the selected attribute.
>
>
> **Q1 "How does the proposed framework handle the issue of defining a cost function for recourse?"**
>
> We consider the definition of cost as an orthogonal, highly application/domain dependent and task, which is out of scope of our work; we refer the reader to the discussion in [30,34] and [A] about the challenges in defining cost functions.
>
> Our framework is built so that it can support arbitrarily defined cost functions on top of the available attributes. Upon this, our framework utilizes the produced recourse scores in different ways (or not at all) depending on the selected definition. In particular though, since we recognize the difficulty of defining costs, two of our definitions actually are cost-oblivious, i.e., they rely on measuring effectiveness of actions and not their costs (making the assumption that the same action will have the same cost for all individuals of the examined subgroup). We note that in our experiments, we select a straightforward cost configuration that is used solely to demonstrate the results of our method, as described in "Experimental Setting" of the supplementary material.
>
>
> **Q2 "Are there any computational or scalability limitations?"**
>
> We note that as reported in the pdf attached to the global rebuttal, the runtime of FACTS is in the order of a few minutes for the datasets examined.
>
>
> **Q3 "How does the performance of the FACTS framework compare to other state-of-the-art approaches for auditing subgroup fairness?"**
>
> In our work, we audit for a specific type of algorithmic fairness, fairness of recourse. All methods of the literature (e.g. Fliptest, Fairtest, "Preventing Fairness Gerrymandering") audit for predictive fairness (like equal opportunity), and thus are not comparable.
>
> Note that mean cost of recourse or "burden" [20,31,36] is the only known fairness of recourse metric in the literature. We also note that AReS [30] and follow-up work are global explainability methods, which can be used for auditing for the burden metric. Such methods however are not designed with auditing as their main goal. In particular, AReS only provides a toy example with qualitative results of how their framework can be used for fairness auditing and states that: "It is thus important to be cognizant of the fact that AReS is finally an explainable algorithm (as opposed to being a fairness technique) that is meant to guide decision makers."
>
> Experiments with real datasets, presented in Tables 1, 6, 7, 8, 9 and also in Tables 11, 13, 15, 17, 19 demonstrate that burden fails to uncover other forms of recourse bias.
>
> **Q4 "the assumption that counterfactual explanations can accurately capture the underlying causes of bias and provide actionable insights."**
>
> Indeed we operate on this assumption. We believe that algorithmic recourse (via counterfactual explanations) is an important aspect of the behavior of a model that captures how difficult it is for an individual to have agency over algorithmic decisions that concern them. In this sense, fairness of recourse mandates that individuals are not discriminated against in their capacity to receive desirable outcomes.
>
>
> **Q5 "The effectiveness of the FACTS framework may vary depending on the specific dataset and domain."**
>
> We note that we have evaluated our method on four widely-used datasets in the fairness literature and we report interesting/meaningful findings for all of them (in the main text and in the supplementary material). We also show that the various fairness of recourse definitions we have introduced are quite distinct notions, and it is thus up to auditors and domain experts to decide how best to apply the FACTS framework.

---

> > ### Comment · Reviewer_EeTT · 2023-08-17
> > **Thank you for your rebuttal**
> >
> > I thank the authors for their efforts in providing a thorough rebuttal. The response addressed most of my concerns. For now, I tend to maintain my score and I look forward to the discussion with other reviewers.

---

### Official Review · Reviewer_fCWq · 2023-07-07

**Soundness:** 3 good
**Presentation:** 3 good
**Contribution:** 3 good
**Rating:** 7
**Confidence:** 3

**Summary:**

This paper considers the problem of fairness of machine learning based decisions. The setting is as follows: there is a set of features X in R^n where X_n denotes the demographic group, and a classifier h: X to {0,1} and we have access to a dataset D of individuals with h(X)=0. Each individual can take actions in an action space A to potentially change their classification decision to h(a(X))=1: this is recourse, and a is the counterfactual explanation. The paper considers how different actions to achieve recourse can be fair on either a micro or macro level.
They first define multiple notions for effectiveness of actions to achieve recourse, that depend on a group, a cost budget or an effectiveness level. They then define and recall multiple definitions to define fairness of recourse (6 such definitions in section 2.3).
They propose an algorithm "FACTS" that enables one to check for subgroups (based on features X) where there exist unfairness based on the 6 definitions and allows one to find the interventions to achieve needed effectiveness rates. They evaluate and showcase their algorithm on four different datasets.

**Strengths:**

Originality: I think the way this paper thinks about recourse is novel in terms of the trade offs between effectiveness, cost and subgroups. Furthermore their algorithm is novel and intuitive way to find violations of fairness of recourse constraints.

Quality: the experimental results seem sound (code is provided) and definitions are well discussed.

Clarity: I enjoyed reading this paper, in particular, the introduction. However, there are some minor changes that can improve the exposition.

Significance: I think the FACTS tool can serve as a nice way to audit algorithms for fairness of recourse. Moreover, the definitions in section 2.3 and the exposition in the introduction is very insightful for the community in algorithmic recourse.

**Weaknesses:**

- there are two limitations of the algorithm: the form of the predicates (subgroups) and the form of the actions. In particular the subgroups are conjunctions of multiple features (does not allow arbitrary subgroups) and second actions are also conjunctions of feature values. This will lead to finding too many subgroups and too many actions, that could have instead been grouped under one subgroup or one action. Second, the algorithm only applies to categorical features and actions.

**Questions:**

- can you address the point in the weaknesses section?

- does the algorithm generalize to multiclass demographic groups and multiclass outcomes?

- how can one find recourse for continuous features based on the provided algorithm?

- the current algorithm lacks statistical tests to check if the subgroup found and the violations are statistically significant, how can one ensure that the violations found are in fact significant?

- what is the runtime of the algorithm and how does it scale when features have very large spaces (i.e. suppose feature X_2 can take 10000 values)

comments:

- I really liked the introduction, but best to separate related work from the introduction. I would also encourage the authors to make the intro more succinct to avoid repeating the explanations in later sections.

- the figures to show the recourse are not easy to read or attractive, I would suggest the authors spend some time improving the design of the CSC (figure 2 and figure 3)

- sections 2.2 and 2.3 now read as a list of definitions without much continuity or story between the definitions and constraints.

**Limitations:**

limitations are adequately addressed.

---

> ### Author Rebuttal · Authors · 2023-08-10
>
> **W-i "There are two limitations of the algorithm: the form of the predicates (subgroups) and the form of the actions. In particular the subgroups are conjunctions of multiple features (does not allow arbitrary subgroups) and second actions are also conjunctions of feature values. This will lead to finding too many subgroups and too many actions, that could have instead been grouped under one subgroup or one action."**
>
> We argue that conjunctions of multiple features is the most natural and interpretable way to define and refer to subpopulations. Investigating algorithmic fairness for clearly defined and understood subpopulations is very important. In contrast, arbitrary definitions of subgroups is also susceptible to gerrymandering, where subgroups may be maliciously defined so as to hide unfairness.
>
> Generating many actions is actually desirable. When we audit for fairness, we consider all of the generated actions, aggregate their effectiveness and cost, and define the effectiveness-cost distribution on which all our fairness metrics are based.
>
>
> **W-ii "Second, the algorithm only applies to categorical features and actions."**
>
> This is not true. There seems to be a misconception about how FACTS handles numerical attributes, which we try to clarify in the author rebuttal. Please refer to the section on *Numerical Attributes and Binning*.
>
> **Q1 "Does the algorithm generalize to multiclass demographic groups and multiclass outcomes"**
>
> The generalization to multiple protected groups (multiclass demographic groups) is straightforward. For example, we can simply compare, say the burden, of each protected group to all others, e.g., one-vs-all. An example of this approach for the Adult dataset and the multi-values race attribute is presented in Figure 3 in the pdf attached to the author response.
>
> Regarding multiclass outcomes, we note that the notion of recourse (and by extension the notion of fairness of recourse) assumes there is a favorable and an unfavorable class, i.e. implies a binary classification setting. This also means that these notions can transfer to a multiclass setting when one of the classes can be considered as the favorable and all others as unfavorable; we note however that we have not seen such a case in the literature of algorithmic recourse.
>
>
> **Q2 "How can one find recourse for continuous features based on the provided algorithm?"**
>
> Please refer to the global rebuttal for a detailed explanation. Briefly the idea is to perform binning on the continuous features and define actions based on these bins.
>
>
> **Q3 "The current algorithm lacks statistical tests to check if the subgroup found and the violations are statistically significant, how can one ensure that the violations found are in fact significant?"**
>
> Regarding the statistical significance of a subgroup/subpopulation, we note that this task should be application and context specific, as when a subgroup should be considered representative might differ in different applications, and even in the same application when different policies are followed. Our framework enables a simple parameterization that lets the auditor decide the sizes of the subpopulations examined (set to 1% of the dataset size in our experiments).
>
> Regarding the statistical significance of fairness violations, we note that for some fairness metrics it is easy to compute the statistical significance. For example, for mean recourse cost, we can simply use the t-test to assess how significant the difference between the means of the two sub-populations is; for fair effectiveness-cost trade-off, we can report the result of the Kolmogorov-Smirnov test, as discussed in the main text.
>
> Nonetheless, we argue that the *importance* of any fairness violation should be best judged by an expert auditor with domain knowledge and assisted by tools such as FACTS. Note that FACTS depicts alongside the unfairness metric (e.g., difference of mean recourse cost), the sub-population sizes (the coverage percentages depicted as blue in Figs. 2 and 3 in the main text).
>
>
> **Q4 "What is the runtime of the algorithm and how does it scale when features have very large spaces (i.e. suppose feature X_2 can take 10000 values)"**
>
> Tables 1 and 3 in the pdf attached to the author rebuttal report the runtime of our algorithm for all datasets, and also includes results while varying the number of bins in the numerical attributes.

---

> > ### Comment · Reviewer_fCWq · 2023-08-16
> > **Response**
> >
> > Thank you for your rebuttal! Raised my score to accept, I recommend accepting this paper.

---

### Official Review · Reviewer_CDVq · 2023-07-08

**Soundness:** 3 good
**Presentation:** 3 good
**Contribution:** 3 good
**Rating:** 6
**Confidence:** 4

**Summary:**

In this paper, the authors explore the fairness of recourse in detail and distinguish between the micro and macro viewpoints. Moreover, they propose an efficient, interpretable, model-agnostic, highly parameterizable framework, called FACTS, to audit for fairness of recourse and provide an interpretable summary of its finding.

**Strengths:**

- The authors claim that their work is the first that audits for fairness of recourse at the subpopulation level.
- The results of their work are explainable and interpretable, which makes it more practical and easier to take action in order to mitigate the bias for different subgroups.
- The authors provide a thorough analysis of various fairness metrics and their advantages and limitations to motivate their work. They also define various subgroup recourse fairness metrics and produce separate subgroup rankings per definition.

**Weaknesses:**

- The experiments section could be improved by including more datasets and comparisons with other fairness metrics and definitions.

**Questions:**

- In general, how realistic is it to be oblivious to the cost function and have an equal choice for recourse?


**Limitations:**

- The experimental section provides limited insight as it only contains one dataset.

---

> ### Author Rebuttal · Authors · 2023-08-10
>
> **W "The experiments section could be improved by including more datasets and comparisons with other fairness metrics and definitions."**
>
> We would like to point out that due to lack of space in the main text, we have only included a single dataset Adult with sex as the protected attribute. In the supplementary material, we have additional experiments using Adult (race is protected), SSL, COMPAS, and Ad campaign datasets. Moreover, in the pdf attached to the author rebuttal, we include results on German credit.
>
> Since fairness of recourse is a quite recent notion of algorithmic fairness, there exist no other fairness metrics beside the average recourse cost or "burden". In results shown in Table 1, and in additional results in the supplementary material, we showcase that the various fairness metrics we introduce capture distinct interpretations of recourse fairness.
>
>
> **Q "In general, how realistic is it to be oblivious to the cost function and have an equal choice for recourse?"**
>
> Equal choice for recourse is one of our proposed definitions of recourse fairness and may find application in some settings. To recall, it says that if you have a set of actions such that each action is considered equal in terms of cost (this can be when actions have equal or comparable cost, or when cost is not an issue or irrelevant or cannot be well quantified) you can be *cost oblivious* and define fairness just in terms of *effectiveness* with respect to this set of actions.
>
> A cost-oblivious setting where equal choice for recourse makes sense can be seen in Figure 1a in our main text. Suppose that a company makes available to its employees two training programs. Action $a_1$ refers to a training program to enhance productivity skills (and thus affect the CTE aspect of individuals), while action $a_2$ refers to a similar training program to enhance project acquisition skills (and thus affect the ACV aspect of individuals). It can be argued that the cost of actions $a_1$ and $a_2$ is irrelevant and that we can be cost oblivious for them. Equal choice for recourse would investigate if individuals from race 0 and race 1 can equally benefit from these actions.

---

> > ### Comment · Reviewer_CDVq · 2023-08-18
> >
> > Thanks for the explanations. I appreciate the authors' efforts to clarify the concerns.

---

### Official Review · Reviewer_m3rQ · 2023-07-10

**Soundness:** 2 fair
**Presentation:** 4 excellent
**Contribution:** 2 fair
**Rating:** 5
**Confidence:** 4

**Summary:**

The paper proposes a framework (termed as FACTS) for analysing the recourse fairness of a machine learning model. The work introduces multiple metrics for quantifying recourse fairness - both at micro level (individuals considered separate) and macro level (individuals considered together). Some proposed recourse fairness metrics are argued to be unaffected by the underlying cost metric. Finally, the work demonstrates the proposed framework for a logistic regression model by training it on several datasets. All except one dataset results are present in the supplementary material.

**Strengths:**

1. Analysing model fairness from the perspective of recourse explanations is an important research problem. There exists much work in analysing the fairness of model predictions, but less on analysing if the generated recourses are fair.
2. The paper is very well-written. The motivation, definitions, and metrics are all clearly explained. The authors cover the current metrics clearly, highlight their weaknesses, and then introduce their proposed metrics.
3. Bibliography is extensive and covers most of the recent papers on the related topics.

**Weaknesses:**

1. Experiments section seems insufficient. Please see below for some examples

   a) The experiments section and the supplementary material presents outputs of the FACTS framework for different datasets, however, it is unclear how to evaluate the efficacy of the proposed framework. It would have been great if the authors had included experiments using toy datasets to create a biased model and then demonstrated that FACTS can uncover the recourse bias while the existing methods don’t/do it with poor coverage.

  b) I think the current experiments could have been more detailed for example by covering other model classes (e.g., tree-based, neural networks) to understand if the proposed metrics generalise to non-linear and non-differentiable models.

  c) FACTS is argued to be scalable, interpretable and highly parametrizable (Line 113). However, there are no experiments to support this. It would be great to understand interpretability through a user study for example. Similarly, analysing how FACTS scales to continuous datasets would be very helpful because as it uses an itemset miming algorithms which generally don't scale well with continuous datasets as feature binning creates a large search space [1].

2. Line 119 - FACTS can explore systematically the feature space. I wonder if there are any guarantees about this? If the feature dimension is high and the dataset has continuous features then the search space becomes too large to scale well.

[1] Dan Ley, Saumitra Mishra, Daniele Magazzeni. Global Counterfactual Explanations: Investigations, Implementations and Improvements. ICLR Workshop 2022.



**Questions:**

1. It seems to create counterfactual summaries, FACTS mines conjunctions of predicates for all positive samples and then uses them to find counterfactuals. Hwoever, doesn't this approach limit the coverage (effectiveness) because if the positive samples are not diverse (i.e., the feature space is not well covered), we may not find counterfactuals for many individuals, resulting in poor coverage.

2. I think FACTS have many similarities in terms of metrics with [2]. For example, micro and macro vs finding a global direction and scaling it locally. Similarly, ECD vs coverage-cost profiles. I would like to hear authors comment on how their work differs from GLOBE-CE.

3. Some points needs clarification/correction
    1. What is "horizontal intervention". lines 342 and line 70
    2. I think the term "effectiveness" (Line 193) is same as the term "coverage" used in the context of GCEs? See Ref[3]
    3. Further, set of actions A can be considered as diverse CFs. ECD profiles same as in GCEs? Micro vs Macro too in G~LOBE-CE
    4. Line 154: I don’t think this claim is correct. AReS[3], GLOBE-CE[2] and Gupta et al. have tried to address this problem before using global (group-level) counterfactuals.
    6. Line 161: It would be helpful to understand authors comments on how FACTS compares to other global explanations framework in terms of efficiency.

[2] A Translation-Based Approach for Global Counterfactual Explanations. Dan Ley, Saumitra Mishra, Daniele Magazzeni. GLOBE-CE: CoRR abs/2305.17021 (2023)

[3] Beyond Individualized Recourse: Interpretable and Interactive Summaries of Actionable Recourses. Kaivalya Rawal and Himabindu Lakkaraju. NeurIPS 2020

**Limitations:**

The authors do not discuss any limitations of their work. However, it is important to note, given that FACTS is proposed as a tool to investigate recourse fairness, detailed experimentation and analysis would be needed before using the tool in the real-world applications.

---

> ### Author Rebuttal · Authors · 2023-08-10
>
> **W1a "experiments using toy datasets to create a biased model and then demonstrated that FACTS can uncover the recourse bias"**
>
> Please note that we have provided a toy example in Fig, 1 where we show how burden is not nuanced enough to capture other notions of recourse bias.
>
> Most importantly, we have experimented with real datasets with known statistical bias, including Adult, COMPAS and IBM Ad Campaign. Tables 1, 6, 7, 8, 9 demonstrate that burden fails to uncover other forms of recourse bias. A more extensive assessment is presented in Tables 11, 13, 15, 17 and 19.
>
> **W1b "covering other model classes (e.g., tree-based, neural networks)"**
>
> We note that our method is model-agnostic and does depend on the model class. Nonetheless, we have considered two additional models, XGBoost and a NN, for the Adult dataset. An example CSC is in Figures 1 and 2 in the pdf attached in the author rebuttal.
>
> **W1c-i "interpretability through a user study"**
>
> This is outside the scope of this work, but we acknowledge the utility of a standardized study to evaluate the interpretability of fairness judgements. We recognize that GLOBE-CE has taken steps towards that direction.
>
> **W1c-ii "analysing how FACTS scales to continuous datasets"**
>
> We present results reporting the runtime as we vary the number of bins per numerical attribute and the minimum support threshold for frequent itemsets. Refer to Tables 3 and 4 in the pdf.
>
> **W2 ""FACTS can explore systematically the feature space." I wonder if there are any guarantees about this. ... the search space becomes too large to scale well."**
>
> FACTS is an offline auditing method. Indeed the search space can become large and this is a known limitation of all algorithms that examine subpopulations. We note that a similar approach is used by FairTest for auditing prediction fairness of subpopulations and by AReS to generate a summary of counterfactual explanations. We also note that GLOBE-CE does not examine subpopulations.
>
>
> **Q1-i "It seems to create counterfactual summaries, FACTS mines conjunctions of predicates for all positive samples"**
>
> We emphasize that FACTS does not aim to create counterfactual summaries. It uses all counterfactuals discovered to audit for fairness.
>
> **Q1-ii "we may not find counterfactuals for many individuals, resulting in poor coverage"**
>
> It is indeed possible that we may not find recourse for some individuals. Note however that this holds for any model-agnostic method that examines a finite set of actions and does not explore all, possibly infinite, actions. The more actions we examine the better the approximation of the effectiveness-cost distribution will be.
> If we observe poor effectiveness (coverage in [A]) in some subpopulations, there are some simple remedies: we may lower the minimum support threshold when mining for actions; we may additionally examine actions generated by a different process, like uniformly sampling the space of possible actions.
>
> **Q2 "I think FACTS have many similarities in terms of metrics with [2]. For example, micro and macro vs finding a global direction and scaling it locally. Similarly, ECD vs coverage-cost profiles. I would like to hear authors comment on how their work differs from GLOBE-CE."**
>
> We summarize the differences:
> - FACTS is a method designed to audit for fairness of recourse. GLOBE-CE aims to find the best counterfactual summary,  which can be used to audit for fairness at the population level.
> - FACTS examines subpopulations and may uncover instances of unfairness hidden at the population level. GLOBE-CE can only summarize the recourses of the entire population.
> - When GLOBE-CE is used to audit for fairness it compares the average cost, aka *burden* in the literature, among the protected subgroups. FACTS is more nuanced, studying in depth the notion of fairness of recourse and making several novel contributions; most notably, it motivates and formalizes a series of fairness metrics, some of which are based on effectiveness and thus oblivious to the cost function, and it defines the micro and macro viewpoints.
> - When GLOBE-CE is used to audit for fairness it draws conclusions based on the best recourse direction discovered, i.e., it forces all individuals to achieve recourse through actions along one direction. In contrast, FACTS draws conclusions by considering all examined actions.
> - "Finding a global direction and scaling it locally" is neither the micro (it restricts the actions available to individuals) nor the macro view (it allows for different actions to individuals).
> - The coverage-cost profile in GLOBE-CE is a *constrained* version of the effectiveness-cost distribution (ECD) in FACTS. Specifically, the coverage-cost profile is an ECD where only actions along a *single* recourse direction are aggregated in the micro viewpoint of FACTS.
>
> **Q3 “Some points needs clarification/correction [...]”**
>
> 1. An intervention for all individuals.
> 2. Correct
> 3. Refer to answer to *Q2*
> 4. Note that the term subgroup/subpopulation refers to a group of individuals defined by more than one attributes, following the terminology in [17]. Regarding the references:
> -AReS [30] is a global explainability method. Quoting “It is thus important to be cognizant of the fact that AReS is finally an explainable algorithm (as opposed to being a fairness technique) that is meant to guide decision makers.” Fairness auditing is merely discussed as a potential merit of the method and not systematically treated.
> -GLOBE-CE [A] is also primarily a global explainability method that can be used for fairness auditing. What is important, is that it does not audit fairness on the subpopulation level.
> - Gupta et al. [10] similarly considers groups defined by a single protected attribute and not subpopulations. Further, the paper focuses on fairness correction and not auditing.
> 5. FACTS is not a global explainability method.

---

> > ### Comment · Reviewer_m3rQ · 2023-08-18
> >
> > Thank you for providing clarifications and sharing more details/experimental results. I have some comments on some rebuttal comments. Please see them below. Further, I summarise my view of the paper towards the end.
> >
> > ***W1c-i "interpretability through a user study"***
> >
> > I appreciate authors acknowledgement of the utility of a user study. Please note that, user studies are important to access if CSCs are helpful and interpretable to an end-user (e.g., an auditor). I suggest to avoid making claims on "interpretability" till we quantitiavely evaluate it. As the authors note, GLOBE-CE performs a user study and so does AReS.
> >
> > ***Q2***
> >
> > - FACTS examines subpopulations and may uncover instances of unfairness hidden at the population level. GLOBE-CE can only summarize the recourses of the entire population.
> >
> > Thanks for the clarification, but I think the above statement for GLOBE-CE is incorrect. Please note that GLOBE-CE aims to find a global direction that translates a given set of data to the desired class. Importantly, the set of data could be entire population or sub-population within it.
> >
> > - When GLOBE-CE is used to audit for fairness it draws conclusions based on the best recourse direction discovered, i.e., it forces all individuals to achieve recourse through actions along one direction. In contrast, FACTS draws conclusions by considering all examined actions.
> >
> > Please note that the above statement is not entirely correct. GLOBE-CE does find a global direction, but providing recourse based on it will result in high cost counterfactuals, hence, the authors propose to scale it locally. This will help to maximise coverage but keeps the cost of recourse low.
> >
> > - The coverage-cost profile in GLOBE-CE is a constrained version of the effectiveness-cost distribution (ECD) in FACTS. Specifically, the coverage-cost profile is an ECD where only actions along a single recourse direction are aggregated in the micro viewpoint of FACTS.
> >
> > Again, I understand, the cost-coverage profile is developed using scaled translation vectors and not only following the global direction. Hence, I believe, ECD and cost-coverage profiles are similar ideas.
> >
> > ***Q3***
> >
> > 1. “Some points needs clarification/correction [...]”
> >
> > Thanks. I would strongly encourage to reuse the existing terms from literature instead of proposing new terms unless there is a need for it. Another example is set of actions A, which goes by the name **diverse counterfactuals** in literature [1][2]. It would be great if authors clarify in the paper these relations.
> >
> > 4. Note that the term subgroup/subpopulation refers.....
> >
> > I understand. Thanks for the clarification. But, as discussed above, GLOBE-CE works at any level (group, subgroup). It is important to clarify this in the paper.
> >
> >
> > **The authors clarified on multiple questions and shared additional results. I think this further clarifies the contribution of this work. I am happy to increase my score. Further, to avoid ambiguity to a reader, I strongly encourage the authors to clarify in the paper how this work differs from other Global CFs methods in the literature.**
> >
> >
> >
> > [1] https://arxiv.org/pdf/1901.04909.pdf
> >
> > [2] https://arxiv.org/abs/1905.07697

---

### Author Rebuttal · Authors · 2023-08-10

We are thankful to the reviewers for their insightful and constructive comments. In this global response, we would like to address some misunderstandings about the focus of our work and how we handle numerical attributes, and also discuss limitations.

### Global Explainability vs Auditing for Fairness

FACTS is a method to audit a model for fairness of recourse in subpopulations. AReS [30, 24] and GLOBE-CE [A] are methods to generate summaries for global counterfactual-based explanations. As such the papers and methods have distinct focus and objectives.

Please note that GLOBE-CE appears in ICML 2023 and was first published on May 26th in arXiv, *after* the NeurIPS 2023 submission deadline of May 8th.

AReS and GLOBE-CE consider global counterfactual explanations (GCEs); a GCE is an explanation that applies to a group of instances collectively (akin to our macro viewpoint). Their methods differ in the GCE definitions. A GCE in AReS is a translation in the feature space (akin to an action in our terminology). A GCE in GLOBE-CE is a direction in the feature space along which instances can achieve recourse with differing translation magnitudes. Nonetheless, AReS and GLOBE-CE have the same goal: find a small set of GCEs to "best" explain a group of instances (according to some objectives). The motivation is to construct an accurate, easily understandable *summary* of counterfactuals; the main contribution of GLOBE-CE is that it constructs a more accurate summary than AReS. Note also the analogy with local explainability, where the goal is to find the best (typically the nearest) counterfactual for a single instance.

In contrast, auditing for fairness is *not* an optimization problem. We do not generate counterfactuals and then select a *few* good among them. In fact, we take the opposite approach: we generate as many counterfactuals as we can afford to, and use *all* of them to approximate the effectiveness-cost distribution.

Global explainability methods can be used to audit for recourse fairness, but only as an afterthought. In contrast, our main objective is to formalize the notion of recourse fairness. We are the first to motivate and define different variants going beyond the mean recourse cost or “burden” in literature, and consider micro and macro viewpoints when examining subpopulations.

Finally, we note that model auditing is typically an offline process, and the runtime is often not an issue. Nonetheless, the runtime of FACTS is in the order of minutes for all datasets and configurations tested, as we report in the pdf attached (Tables 1 and 3).

### Numerical Attributes and Binning

We would like to clarify that FACTS indeed works for datasets/models with numerical/continuous features; in fact it works with any mixture of categorical and numerical features. The core idea is that numerical attributes are binned. These bins are used to define subpopulations, e.g., people in the salary range [40K,50K], and actions, e.g., make salary [50K,60K]. So the action "if salary is [40K,50K], make salary [50K,60K]" means that all individuals within that salary range are mapped to their counterfactuals where their salaries are increased by 10K.

Binning is *necessary* when defining subpopulations, as we want to explore the entire feature space and present conclusions in a manner that is easily understandable by humans. E.g. compare the interpretability of "there is unfairness against females when married=no, salary in [40K,50K]" with that of "... when married=no, salary=39K or 42K or 45.5K".

Binning is also *necessary* when considering actions over numerical attributes. As explained, binning of granularity 10K for salary means that we consider actions that change salary by ±10K, ±20K, etc. We emphasize that because the number of actions/counterfactuals is infinite, *all* methods that work with model-agnostic counterfactual-based explanations (e.g., FACTS, AReS, GLOBE-CE) have to explore only a small set of actions. Note again the distinction between global explainability and auditing for fairness. In global explainability only a few of the explored actions will be selected, whereas we argue that all of them should be used to reliably audit for fairness.

Recall that our goal is to approximate the effectiveness-cost distribution, which means we should consider as many actions as possible. Therefore, on the one hand, binning for actions should rather be fine-grained, while on the other hand, binning for subpopulations should be moderately coarse-grained so as to draw conclusions that concern many affected individuals.

With that said, the binning granularity for actions and subpopulations can *differ*. For example consider bins of length 10K for subpopulations and 5K for actions. An action, "if salary is [40K,50K], make salary [55K,60K]" means that individuals with salary in [40K,45K] increase their salary by 15K, and individuals within [45K,50K] increase their salary by 10K.

In all our experiments, we have shown results using the same binning for actions and subpopulations. Adapting our algorithm to handle differing binning granularities is straightforward. Note that Tables 3 and 4 in the attached pdf present results as we vary the binning granularity.

### Limitations and Potential Societal Impact

We would like to thank the reviewers for raising awareness on several aspects, which we intend to discuss in the suppl. material and include in the guidelines for ethical usage of FACTS:
- the definition of costs to actions
- transparency about the subpopulations explored
- how to interpret fairness metrics and auditing results (incl. importance and statistical significance)
- need for a user study on interpretability
- compliance with GDPR and similar regulations in auditing using protected/sensitive attributes
- discussion and study on robustness and sensitivity to malicious actors.


[A] GLOBE-CE: A Translation-Based Approach for Global Counterfactual Explanations. Ley et al. ICML 2023.

---

### Decision · Program_Chairs · 2023-09-21

**Decision:**

Accept (poster)

**Comment:**

After some discussion, the reviewing team felt that this work had enough merits to accept it as a poster.